# RAR: Reversing Visual Attention Re-Sinking for Unlocking Potential in Multimodal Large Language Models

**Zhehan Kan**[1,2]* **Xin Li**[2]* **Yanli Liu**[1]* **Xiaochen Yang**[3] **Xinghua Jiang**[2]
**Yinsong Liu**[2] **Deqiang Jiang**[2] **Xing Sun**[2] **Qingmin Liao**[1] **Wenming Yang**[1]†
[1]Tsinghua University  [2]Tencent Youtu Lab  [3]The University of Glasgow

## Abstract

Multimodal Large Language Models (MLLMs) have achieved remarkable success in vision-language tasks, yet they frequently exhibit suboptimal output layers, where intermediate decoder layers outperform the final ones, signaling underutilized model capacity. In this work, we delve into the root causes and attribute this issue to the ***Visual Attention Re-sinking*** phenomenon, precipitated by attention gradient sparsity driven by textual supervision dominance. This degradation causes attention heads to evolve into sink heads that prioritize low-semantic backgrounds, thereby disrupting modality fusion, neglecting visual information, and biasing outputs toward textual priors, ultimately impairing model performance. To mitigate this, we introduce a parameter-free ***Sink Attention Dynamic Sparsification (SADS)*** framework that dynamically preserves all vision heads, ensuring focused attention on semantically salient regions, while retaining only a minimal subset of sink heads, including a designated shared head to safeguard essential global and contextual information. Integrated into diverse MLLMs, our framework yields substantial performance gains across 20 benchmarks spanning five task categories (visual grounding, general VQA, OCR-related VQA, vision-centric tasks, and visual hallucination mitigation) surpassing supervised fine-tuning while boosting inference speed by 10.3%. This approach offers a novel avenue for maximizing MLLMs capabilities.

## 1 Introduction

In recent years, Multimodal Large Language Models (MLLMs) have rapidly evolved into a scalable pathway toward Artificial General Intelligence (AGI) (Bai et al., 2025; Chen et al., 2024; Liu et al., 2023). These models typically employ a vision encoder and a connector to align image features with text embeddings, which are then processed by an LLM decoder to generate responses (Bai et al., 2025; Chen et al., 2024; Liu et al., 2023). While MLLMs excel in vision-language tasks like visual question answering, grounding, and captioning, recent findings show mid-to-late vision encoder layers often surpass the output layer, due to CLIP training fostering rich spatial and semantic features in intermediates (Bolya et al., 2025). Analogously, for hallucination mitigation in MLLM decoders, mid-layer visual facts are suppressed later, leading to methods that leverage or fuse intermediates for enhanced outputs (Wang et al., 2024). However, existing research offers limited insights into the underlying causes and primarily relies on post-hoc remedial strategies that fail to fully activate the model's capacity. Thus, addressing ***"why the output layer in MLLMs is suboptimal"*** and ***"how to maximize MLLM capabilities by optimizing the output layer"*** represents a critical and urgent challenge.

In this work, we investigate why MLLM output layers are suboptimal. Unlike LLMs, MLLMs must fuse vision and language, typically processing features in early layers, aligning modalities in mid-layers, and organizing responses in late layers (Zhang et al., 2025). As illustrated in Figure 1, we discover that supervision in existing MLLM training paradigms is entirely textual, devoid of direct

---

*Equal contribution. Zhehan Kan works done during the internship at Tencent Youtu Lab.
†Corresponding author.

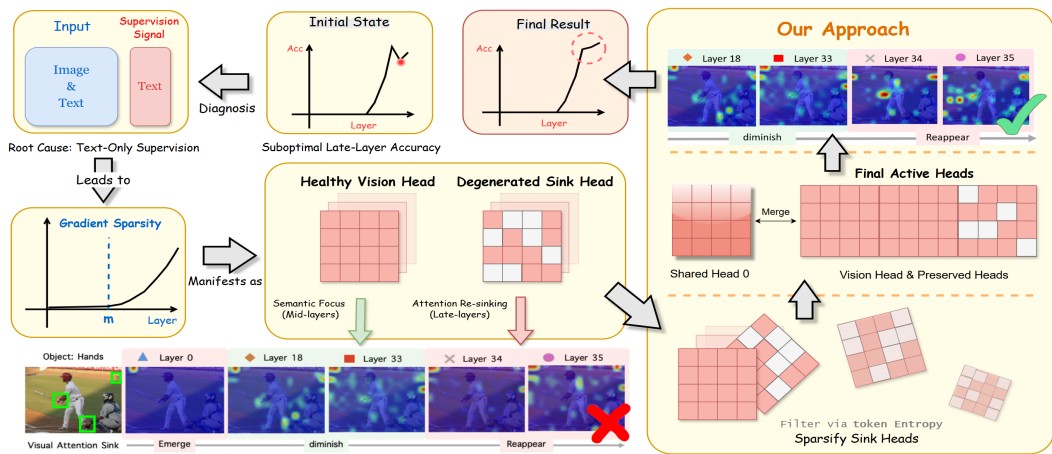

Figure 1: Overview of visual attention re-sinking in MLLMs caused by text-only supervision, inducing gradient sparsity, head degeneration, and suboptimal late-layer accuracy. Our SADS framework sparsifies sink heads while retaining all vision heads and a shared head, achieving progressive accuracy gains and eliminating re-sinking.

visual oversight. Consequently, gradients for vision tokens rely solely on backpropagation of textual losses through the attention mechanism, constraining its learning capacity on vision tokens and rendering the overall gradient distribution increasingly sparse. This sparsity prompts the model, in subsequent forward passes, to concentrate visual attention weights on a shrinking subset of vision tokens. As training iterations advance, these weights progressively localize to an even smaller number of tokens, engendering the **_Visual Attention Re-sinking_** phenomenon, wherein visual attention in late layers reverts to low-semantic backgrounds. This disrupts the modality fusion balance established in mid-layers, compelling the model to increasingly rely on textual priors rather than deeply integrating visual cues. The degradation escalates with iterations, propagating backward from late to mid-layers and ultimately culminating in suboptimal output-layer performance.

Building upon these insights, we introduce a parameter-free **_Sink Attention Dynamic Sparsification (SADS)_** framework. This approach dynamically retains all vision heads during inference while preserving only a minimal subset of sink heads, thereby encouraging the model to prioritize visual information without sacrificing critical global and contextual knowledge. Specifically, we observe that the maximum visual attention between vision and sink heads follows a bimodal Gaussian distribution. Similarly, within the sink heads, the entropy of non-vision token cross-attention between heads that focus on global knowledge (denoted as $sink_G$) and those that fully sink attention onto specific non-fixed tokens (denoted as $sink_S$) also adheres to a bimodal Gaussian distribution. By leveraging the valley of this distribution as a dynamic threshold, our framework retains all vision heads along with the $sink_G$ heads, while designating the first head as a shared sink to ensure the preservation of global and contextual information. This framework is architecture-agnostic and readily applicable to diverse MLLMs. In this study, we integrate it into Qwen2.5-VL, InternVL2, and LLaVA-1.5 (Bai et al., 2025; Chen et al., 2024; Liu et al., 2023). Extensive experiments across a variety of tasks demonstrate that fine-tuning with our framework achieves substantial improvements over standard supervised fine-tuning (SFT) on numerous visual benchmarks. Furthermore, by streamlining redundant attention weight computations, inference speed is boosted by 10.3%.

In summary, our contributions are as follows: 1) We conduct an in-depth investigation into the causes of suboptimal MLLM output layers, attributing them to the text-only supervision paradigm in MLLMs, which, as training iterates, prompts models to learn modality-irrelevant output shortcuts, sparsifying attention gradients, inducing visual attention re-sinking, leading to suboptimal outputs. 2) We introduce a SADS framework that addresses the visual attention re-sinking issue, optimizes attention gradient sparsity, and achieves output-layer optimality, thereby maximally activating the model's capacity. 3) We validate the superior effectiveness and inference efficiency of our method through comprehensive experiments and analyses across five task categories: visual grounding, general VQA, OCR-related VQA, vision-centric tasks, and visual hallucination tasks, spanning 20 benchmarks, providing a novel framework for advancing MLLMs.

## 2 RELATED WORK

**Best Layer in MLLM Decoders.** In MLLMs, the vision encoder's final layer typically extracts visual features, with the decoder's output layer generating responses. However, studies show mid-to-late vision encoder layers often outperform the final one across tasks (Bordes et al., 2022; Chen et al., 2020b; Ma et al., 2024; Shekhar et al., 2023; Walmer et al., 2022; Zheng et al., 2016). For example, iGPT's intermediate layers excelled in image classification (Chen et al., 2020a), and PE demonstrated CLIP training fosters rich features in intermediates across encoders (Bolya et al., 2025). Similar suppression of mid-layer visual facts occurs in MLLM decoders for hallucination mitigation, with fusion methods improving outputs (Wang et al., 2024; Huang et al., 2024). Yet, prior work lacks deep causal insights and relies on post-hoc fixes. In contrast, we attribute suboptimal outputs to visual attention re-sinking from textual supervision and gradient sparsity, proposing SADS for optimal output layers and capacity maximization.

**Visual Attention Sink in MLLMs.** In LLMs, attention sink involves low-semantic tokens (e.g., BOS, ".") drawing excessive weights (Xiao et al., 2023), minimally contributing to inference (Bondarenko et al., 2023). Recent views frame it as first-token mechanistic mixture suppression to prevent collapse (Barbero et al., 2025). In MLLMs, visual attention targets image patches (Aflalo et al.), but often misallocates to low-semantic areas, mitigated by registration tokens (Darcet et al., 2023) or boosted image weights (Zhu et al., 2025). VAR formalized visual attention sink, linking it to sink token activations like LLMs, and reallocating attention (Kang et al., 2025). Conversely, we pioneer the discovery of visual attention re-sinking in MLLMs.

## 3 PRELIMINARIES

MLLMs typically feature an end-to-end architecture integrating a vision encoder, a projection module, and an LLM decoder (Liu et al., 2023; Bai et al., 2025). The vision encoder extracts hierarchical visual features from input images, which are projected into a modality-aligned latent space to bridge visual-textual gaps. These visual embeddings are concatenated with tokenized system prompts and instructions, forming a unified sequence fed into the LLM decoder for autoregressive response generation with causal masking. Each sequence element is a discretized token embedding.

Formally, let $\mathbf{v} \in \mathbb{R}^{N_v \times d}$ denote $N_v$ visual tokens and $\mathbf{t} \in \mathbb{R}^{N_t \times d}$ denote $N_t$ textual tokens ($d$: embedding dimension). The concatenated input $\mathbf{x} = [\mathbf{v}; \mathbf{t}] \in \mathbb{R}^{(N_v + N_t) \times d}$ is processed by the Transformer-based decoder. Each of $L$ blocks computes:

$$\hat{\mathbf{h}}^\ell = \text{LayerNorm}(\mathbf{h}^{\ell-1} + \text{MHA}(\mathbf{h}^{\ell-1})), \quad \mathbf{h}^\ell = \text{LayerNorm}(\hat{\mathbf{h}}^\ell + \text{FFN}(\hat{\mathbf{h}}^\ell)), \tag{1}$$

where $\mathbf{h}^{\ell-1} \in \mathbb{R}^{N \times d}$ ($N = N_v + N_t$, $\mathbf{h}^0 = \mathbf{x}$), LayerNorm is layer normalization (Ba et al., 2016), and FFN is a two-layer feed-forward network with non-linear activation. The multi-head attention (MHA), key to modality fusion, is:

$$\text{MHA}(\mathbf{Q}, \mathbf{K}, \mathbf{V}) = \text{Concat}(\text{head}_1, \ldots, \text{head}_H)\mathbf{W}^O, \tag{2}$$

with $\mathbf{Q} = \mathbf{h}^{\ell-1}\mathbf{W}^Q$, $\mathbf{K} = \mathbf{h}^{\ell-1}\mathbf{W}^K$, $\mathbf{V} = \mathbf{h}^{\ell-1}\mathbf{W}^V$ ($\mathbf{W}^Q, \mathbf{W}^K, \mathbf{W}^V, \mathbf{W}^O \in \mathbb{R}^{d \times d}$). Each head $i$ is:

$$\text{head}_i = \text{softmax}\left(\frac{\mathbf{Q}\mathbf{W}_i^Q(\mathbf{K}\mathbf{W}_i^K)^\top}{\sqrt{d_k}} + \mathbf{M}\right)(\mathbf{V}\mathbf{W}_i^V), \tag{3}$$

where $d_k = d/H$, $H$ is the number of heads, and $\mathbf{M}$ is the causal mask ($\mathbf{M}_{i,j} = 0$ if $j \leq i$, else $-\infty$) (Vaswani et al., 2017). We focus on cross-modal interactions, specifically textual queries attending to visual keys via the visual attention matrix:

$$\text{Visual Attention} = \text{softmax}\left(\frac{\mathbf{Q}_t\mathbf{K}_v^\top}{\sqrt{d_k}}\right) \in \mathbb{R}^{N_t \times N_v}, \tag{4}$$

where $\mathbf{Q}_t$ and $\mathbf{K}_v$ are textual queries and visual keys, analyzed for fusion patterns.

# 4    WHY IS THE OUTPUT LAYER NOT OPTIMAL?

As illustrated in Figure 2, advanced open-source MLLMs consistently face the challenge of suboptimal output layers, where intermediate decoder layers outperform the final layer. This indicates that the extensive parameter capacity underpinning these models is not fully activated. In this section, we investigate the underlying causes of this phenomenon.

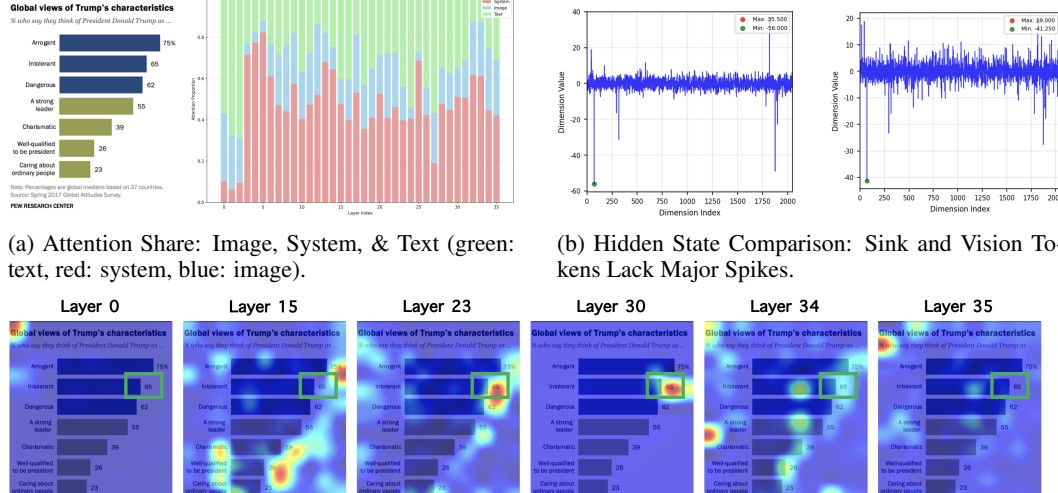

(a) Attention Share: Image, System, & Text (green: text, red: system, blue: image).

(b) Hidden State Comparison: Sink and Vision Tokens Lack Major Spikes.

(c) Evolution of visual attention across layers, where the visual attention sink appears in early layers, diminishes in middle layers, and re-emerges in later layers. Green boxes indicate semantically relevant visual regions.

Figure 3: Analysis of visual attention on the object "the number of intolerant".

## 4.1    VISUAL ATTENTION RE-SINKING LEADS TO SUBOPTIMAL OUTPUT LAYERS.

In LLM decoders, intermediate and output layers perform comparably on simple tasks, while deeper output layers excel on more complex ones, without notable suboptimal output layer issues (Fan et al., 2024). Therefore, we focus our investigation on the fusion of visual and linguistic modalities. Within Transformer-based decoder architectures, models rely more heavily on attention mechanisms than FFNs to inject visual information into the linguistic latent space for modality alignment (Vaswani et al., 2017). Thus, our analysis centers on visual attention. We decompose the impact of visual at-

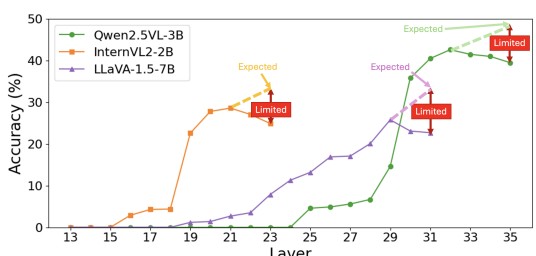

Figure 2: Cross-layer accuracy of the three base models on the OVDEval test benchmark.

tention into two aspects: 1) *the total attention allocated to images* and 2) *the distribution of visual attention across vision tokens*. As shown in Figure 3a, we first compute the attention distribution across image, system, and text components, observing a stable pattern across layers without fluctuations in image attention allocation in later layers. This suggests that the total attention allocated to images is not the primary cause of suboptimal output layers, implying that the issue likely lies in the distribution of visual attention across different vision tokens. According to Equation 4, we compute the attention weights between output tokens and vision tokens to derive visual attention maps. As illustrated in Figure 3c, we observe that visual attention in early layers predominantly concentrates on low-information background regions; transitions to semantically salient areas in mid-layers; and reverts to low-information backgrounds in late layers. We define tokens attracting visual attention to low-information background regions as *sink tokens*, those focusing on semantically relevant regions as *vision tokens*, and the resurgence of visual attention toward low-information backgrounds in late

layers as the ***visual attention re-sinking*** phenomenon. We posit that this re-sinking contributes to performance degradation. To validate this hypothesis, we perform a training-free intervention on the $VQA^{vg}$ test set, reallocating attention weights from sink tokens in the last five layers to semantically pertinent vision tokens, which yields a 0.74% accuracy improvement. This confirms that visual attention re-sinking is the primary culprit behind suboptimal output layers. Accordingly, our investigation focuses on late layers to address the question: ***"What causes visual attention re-sinking?"***

## 4.2 ATTENTION GRADIENT SPARSITY LEADS TO VISUAL ATTENTION RE-SINKING

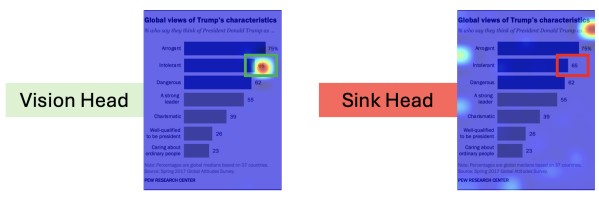

(a) Visual attention of sink head and vision head. Boxes in the images indicate semantically relevant visual regions.

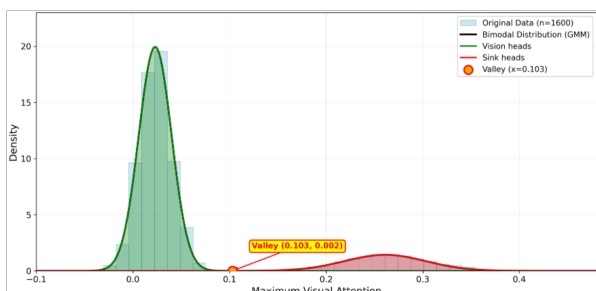

(b) Distribution of maximum visual attention for vision heads and sink heads, showing a bimodal distribution.

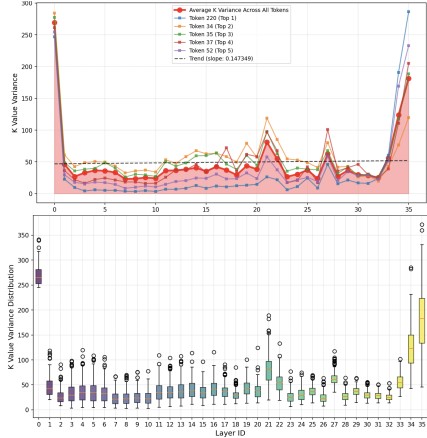

(c) Evolution of visual attention K-value variance in sink heads across layers, exhibiting a decline-plateau-sharp increase trend; fluctuations caused by outlier sink tokens, with boxplots showing more sink tokens in early and late layers.

Figure 4: Analysis of vision and sink heads visual attention on the object "the number of intolerant".

Unlike the attention sink in LLMs and VAR, which stems from massive activations in specific hidden state dimensions and appears in early layers while vanishing in later ones, as depicted in Figure 3b and Figure 3c, we observe that visual attention sink emerges in early layers, diminishes in mid-layers, and reappears in late layers, without massive activations in specific hidden state dimensions. Therefore, we investigate the characteristics of these sink tokens and the reasons for their resurgence.

Table 1: Analysis of the impact of heads on OVDEval benchmark.

| Method | Accuracy (%) |
|---|---|
| Qwen2.5-VL-3B | 39.5 |
| w/o $sink_S$ head | 43.8 |
| w/ 1 $sink_S$ head | 43.0 |
| w/o 1 $sink_G$ head | 43.2 |
| w/o 1 vision head | 42.6 |

**Sink Tokens Concentrate in Sink Heads.** For multi-head attention (MHA), we begin by examining the visual attention representations across individual heads. As illustrated in Figure 4a, we identify pronounced disparities among heads in terms of visual attention allocation. We classify heads that direct visual attention toward semantically salient positions as *vision heads*, whereas those gravitating toward low-information background regions are termed *sink heads*. Notably, the maximum visual attention values in vision heads substantially exceed those in sink heads. To substantiate this observation, we perform a statistical analysis of the maximum visual attention across 1,600 heads in late layers, revealing a distinct bimodal distribution: sink heads cluster at significantly lower maximum visual attention levels compared to vision heads. This pattern yields two key insights: 1) *sink tokens predominantly concentrate within sink heads*, and 2) *although sink tokens exhibit high relative attention weights, their absolute magnitudes remain low*. Consequently, our subsequent analysis focuses on sink heads.

As depicted in Figure 5, we compute the cross-attention among all tokens within sink heads and observe distinct patterns in the non-vision token cross-attention: some heads exhibit uniformly dispersed attention across tokens (preserving global and contextual information), while others sink

attention onto individual low-semantic tokens. Specifically, we quantify this pattern via the entropy of the non-vision token attention distribution, defined as follows. Let $\mathbf{A} \in \mathbb{R}^{L_q \times L_k}$ denote the attention matrix for a given head, where $L_q$ and $L_k$ are the query and key sequence lengths, respectively. Let $\mathcal{I}$ be the set of indices corresponding to non-vision tokens (keys). Extract the submatrix $\mathbf{A}_{\text{sub}} = \mathbf{A}[:, \mathcal{I}] \in \mathbb{R}^{L_q \times |\mathcal{I}|}$. Renormalize each row $i$ of $\mathbf{A}_{\text{sub}}$ such that $\mathbf{A}_{\text{sub}}[i, :] \leftarrow \mathbf{A}_{\text{sub}}[i, :] / \sum_{j \in \mathcal{I}} \mathbf{A}_{\text{sub}}[i, j]$, ensuring each row sums to 1 over non-image keys. Then, compute the average attention distribution $\mathbf{p} \in \mathbb{R}^{|\mathcal{I}|}$ where $p_j = \frac{1}{L_q} \sum_{i=1}^{L_q} \mathbf{A}_{\text{sub}}[i, j]$ for each $j \in \mathcal{I}$. The entropy is given by $H = -\sum_{j \in \mathcal{I}} p_j \log p_j$.

We define high-entropy heads, which attend to global and contextual knowledge, as $sink_G$ heads, and low-entropy heads, which sink attention onto individual low-semantic tokens, as $sink_S$ heads. This complete sinking of attention onto low-information tokens disrupts effective modality fusion, marginalizes visual cues, and biases outputs toward textual priors, ultimately leading to degraded model performance. To validate this, as shown in Table 1, on the OVDEval benchmark, adding an extra $sink_S$ head or ablating vision heads or $sink_G$ heads leads to significant performance drops. These findings highlight the adverse effects of attention sinking to isolated tokens while confirming the importance of visual cues and global context. Moreover, we observe that the entropy values across heads follow a bimodal Gaussian distribution, allowing us to leverage its valley as a dynamic threshold for differentiating $sink_S$ and $sink_G$ heads.

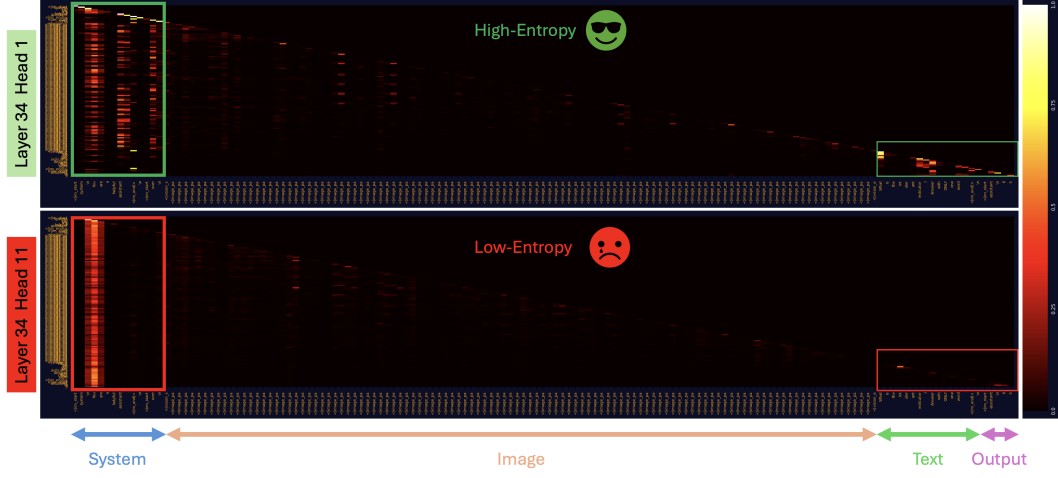

Figure 5: **Attention heatmaps of two sink heads.** Top: High-entropy $sink_G$ head (green box) with distributed attention preserving global context. Bottom: Low-entropy $sink_S$ head (red box) sinking to specific low-semantic tokens.

**Re-sinking Tokens Exhibit High Key Variance.** On sink heads, we concentrate on the key matrix $\mathbf{K}$ in multi-head attention due to its role in modality interactions. As illustrated in Figure 4b, we discover that $\mathbf{K}$ value variances align with the visual attention re-sinking pattern: decreasing in early layers, stabilizing in mid-layers, and surging in late layers. From the box plots, mid-layer fluctuations arise from outlier tokens, which are precisely sink tokens. In early and late layers, the increasing number of sink tokens elevates overall $\mathbf{K}$ variances. It is this high dimensional variance in $\mathbf{K}$ that leads to higher attention weights for sink tokens compared to others.

**Attention Gradient Sparsity Causes Sink Token Resurgence.** Given that supervision in the final layer of MLLMs during training is entirely textual, devoid of direct oversight for visual signals, the gradients for visual tokens rely exclusively on backpropagation of textual losses through the attention mechanism. This dependency constrains the learning capacity of the attention mechanism on visual tokens, rendering the overall gradient distribution increasingly sparse. Consequently, in subsequent forward passes, the model tends to concentrate visual attention weights on a diminishing subset of vision tokens. As training goes on, these weights slowly focus on fewer tokens, eventually forming the "visual attention re-sinking" phenomenon. To test this cause-and-effect link with data, as shown in Figure 7, we track the change in attention gradient sparsity and the number of sink heads during the training process. Notably, at about 2,000 iterations, gradients in late layers start to thin

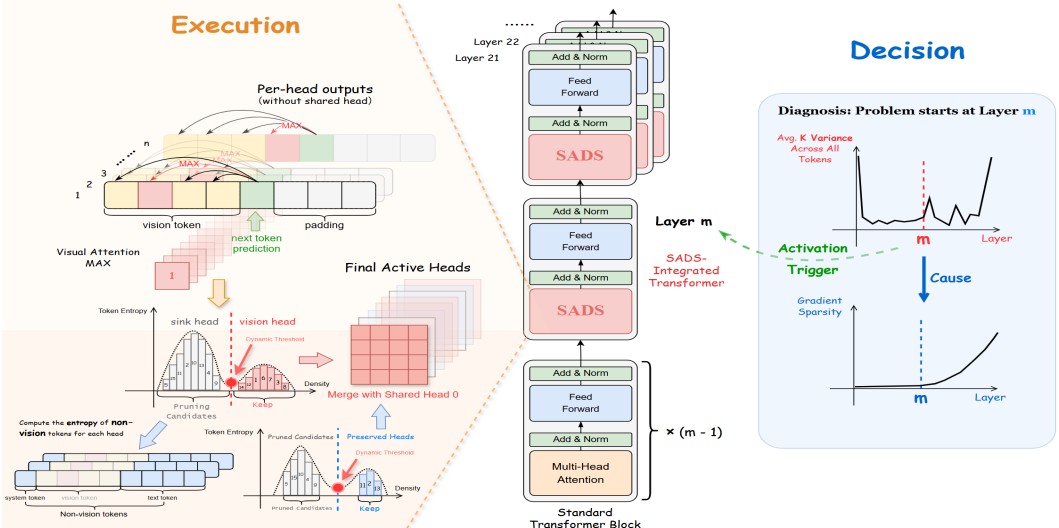

Figure 6: **Overview of the SADS framework. Decision (Right):** Triggers activation at Layer $m$ upon detecting gradient sparsity and key variance anomalies. **Execution (Left):** Filters heads using bimodal thresholds on maximum visual attention (separating vision and sink heads) and non-vision token entropy (distinguishing $sink_G$ from $sink_S$ within sinks). Merges all vision heads, retained $sink_G$ heads, and a shared head for computation, preserving global context.

out; by about 3,000 iterations, the number of sink heads in these layers begins to grow, confirming that this sparsity causes the sink tokens to return.

## 5 SINK ATTENTION DYNAMIC SPARSIFICATION FRAMEWORK

Drawing upon the preceding analysis, we conclude that MLLMs require a sufficient number of vision heads to effectively process dense, semantically rich visual information, complemented by a minimal subset of sink heads to handle textual and global contextual elements. To this end, we introduce the Sink Attention Dynamic Sparsification (SADS) framework. As illustrated in Figure 6, this framework first distinguishes vision heads from sink heads via the bimodal distribution of maximum visual attention. It then retains all vision heads while dynamically sparsifying the sink heads: leveraging the bimodal Gaussian distribution of non-vision token cross-attention entropy, it identifies and preserves the high-entropy $sink_G$ heads (capture global and contextual knowledge) using the distribution's valley as a dynamic threshold, and designates the first head as a shared sink to ensure model stability and safeguard essential global information.

### 5.1 SINK HEAD IDENTIFICATION

Building on the observation from Section 4.2 that the maximum visual attention in vision heads far exceeds that in sink heads, we model each layer in SADS using a Gaussian bimodal distribution, formalized as: $p(x) = \sum_{k=1}^{2} \pi_k \mathcal{N}(x \mid \mu_k, \sigma_k^2)$, where the valley $\alpha = \arg\min_x p(x)$ serves as the maximum visual attention threshold for classifying sink and vision heads. Subsequently, within the identified sink heads, we further delineate them based on the entropy of non-vision token cross-attention, which exhibits a similar

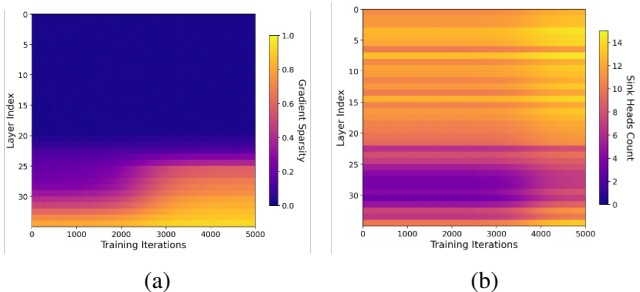

(a)        (b)

Figure 7: (a) The evolution of attention gradient sparsity across layers over training iterations during training. (b) The evolution of sink heads numbers across layers over training iterations during training.

bimodal Gaussian distribution. We analogously model this entropy distribution as $p(e) = \sum_{k=1}^{2} \pi_k \mathcal{N}(e \mid \mu_k, \sigma_k^2)$, with the valley $\beta = \arg\min_e p(e)$ acting as the threshold to distinguish high-entropy $sink_G$ heads (focusing on global and contextual knowledge) from low-entropy $sink_S$ heads (sinking attention onto isolated low-semantic tokens). This approach leverages Gaussian Mixture Models (GMMs) to capture the inherent separation, ensuring robust identification. We fit the GMM using expectation-maximization, which converges efficiently and provides probabilistic assignments, enhancing reliability in noisy attention distributions.

## 5.2 SINK ATTENTION DYNAMIC SPARSIFICATION

Leveraging the precise identification of sink heads, as illustrated in Figure 6, we introduce the parameter-free SADS framework. This approach adaptively retains all vision heads per layer while dynamically sparsifying the sink heads: it preserves the high-entropy $sink_G$ heads (capturing global and contextual knowledge) based on the valley of the bimodal entropy distribution as a threshold, and designates the first head as a shared sink to handle essential textual and global information. Informed by the analysis in Section 4.2 regarding visual attention re-sinking in later layers, we activate SADS from layers displaying variance fluctuations in attention keys and gradient sparsity. This selective activation maintains early-layer modality alignment while enhancing efficiency in later layers. During fine-tuning, SADS compels the model to prioritize visual features, thereby averting textual shortcuts and promoting deeper vision-text fusion, as evidenced by reduced hallucinations and improved visual grounding in downstream tasks.

## 6 EXPERIMENTS

### 6.1 EXPERIMENTAL SETUP

**MLLMs.** In this work, we employ Qwen2.5-VL-3B, Qwen2.5-VL-7B, Qwen2.5-VL-32B, InternVL2-2B, and LLaVA-1.5-7B (Bai et al., 2025; Chen et al., 2024; Liu et al., 2023) as our base models. Notably, as a parameter-free method, SADS can be readily applied to various MLLMs.

**Tasks and Evaluation Benchmarks.** We evaluate our method across a broad spectrum of tasks, categorized into: visual grounding tasks, general VQA tasks, OCR-related VQA tasks, vision-centric tasks, and visual hallucination tasks. 1) Visual grounding tasks assess MLLMs' visual localization capabilities, including Referring Expression Comprehension (REC): RefCOCO/g/+, LISA, RefGTA, and Object Detection (OD): COCO and OVDEval (Lin et al., 2014; Lai et al., 2024; Yao et al., 2024). 2) General VQA tasks evaluate MLLMs' comprehension of image-text pairs, encompassing VQA_v2, VizWiz, VQA_vg, GQA, MME, MMB, MMStar, and AI2D (Jia et al., 2024; Gurari et al., 2018; Krishna et al., 2016; Ainslie et al., 2023; Fu et al., 2024; Liu et al., 2024; Chen et al., 2024; Kembhavi et al., 2016). 3) OCR-related VQA tasks gauge MLLMs' proficiency in high-granularity recognition for icon-document pairs, such as InfoVQA, TextVQA, and DocVQA (Mathew et al., 2021b;a; Singh et al., 2019). 4) Vision-centric tasks focus on visual-spatial perception in image-text understanding, including MMVP, CVBench, and CLEVER Tong et al. (2024a;b); Johnson et al. (2016). 5) Visual hallucination tasks measure the authenticity and reliability of MLLM outputs, featuring POPE and CHAIR (Sun, 2025; Li et al., 2023).

**Training Datasets and Implementation Details.** Across the five task categories, we aggregate a total of 670k training samples sourced from RefCOCO, Dcube, VG, GQA, OCR-VQA, Text-VQA, and CLEVER for model fine-tuning. Our hyperparameters remain consistent with those of SFT and the base models across all benchmarks. For layer selection in SADS, we base our choices on the layers exhibiting K variance fluctuations and the onset of gradient sparsity: for Qwen-2.5VL-3B, we commence from layer 20, for InternVL2-2B, from layer 15, for LLaVA-1.5-7B, from layer 20.

### 6.2 QUANTITATIVE RESULTS

Tables 2 and Tables 3 sequentially present the performance of RAR across the five task categories on 20 benchmarks using three base models. It is evident that RAR consistently outperforms both SFT and the base models on all benchmarks, demonstrating enhanced capabilities in visual localization, visual understanding, spatial perception, and hallucination mitigation. This underscores the effectiveness, scalability, and robustness of the RAR method. Notably, on out-of-distribution

Table 2: Benchmark performance comparison on general VQA and OCR VQA tasks.

| Model | General VQA Task | | | | | | | OCR VQA Task | | |
|---|---|---|---|---|---|---|---|---|---|---|
| | $VQA^{v2}$ | GQA | $VQA^{vg}$ | MME | MMB | MMStar | AI2D | InfoVQA | TextVQA | DocVQA |
| LLaVA-1.5-7B | 78.3 | 61.1 | 54.6 | 1808.4 | 61.1 | 33.2 | 55.7 | 41.2 | 64.7 | 69.4 |
| +SFT | 79.1 | 63.1 | 55.7 | 1899.6 | 61.9 | 34.5 | 56.2 | 43.7 | 65.5 | 71.2 |
| **+ Ours** | **80.8** | **65.2** | **58.5** | **2018.8** | **63.2** | **36.3** | **57.1** | **46.3** | **67.5** | **74.4** |
| Qwen2.5VL-3B | 76.7 | 60.4 | 54.3 | 2184.1 | 75.4 | 53.0 | 77.9 | 75.1 | 78.7 | 93.0 |
| +SFT | 77.9 | 62.0 | 55.2 | 2199.9 | 75.9 | 53.7 | 78.4 | 75.9 | 79.0 | 92.9 |
| **+ Ours** | **79.7** | **64.2** | **58.1** | **2276.3** | **76.9** | **55.4** | **79.5** | **77.3** | **80.4** | **93.5** |
| InternVL2-2B | 72.9 | 55.6 | 50.1 | 1864.3 | 69.1 | 48.9 | 73.1 | 58.8 | 73.4 | 86.4 |
| +SFT | 74.2 | 56.9 | 52.3 | 1899.1 | 70.0 | 49.6 | 73.9 | 59.1 | 73.8 | 86.6 |
| **+ Ours** | **75.9** | **59.0** | **55.4** | **2006.5** | **71.6** | **50.8** | **75.7** | **60.5** | **75.9** | **88.2** |
| Qwen2.5VL-7B | 81.6 | 65.8 | 60.5 | 2276.3 | 82.2 | 64.2 | 84.1 | 81.7 | 80.2 | 94.8 |
| +SFT | 81.9 | 66.1 | 61.0 | 2230.2 | 82.0 | 64.5 | 84.4 | 82.0 | 80.7 | 94.2 |
| **+ Ours** | **82.6** | **67.9** | **62.1** | **2289.8** | **83.3** | **66.0** | **84.8** | **82.9** | **81.3** | **95.0** |
| Qwen2.5VL-32B | 82.9 | 68.4 | 63.6 | 2297.4 | 83.8 | 70.3 | 85.2 | 83.4 | 82.8 | 94.8 |
| +SFT | 83.0 | 68.6 | 63.9 | 2255.4 | 83.8 | 69.6 | 85.1 | 83.0 | 82.9 | 94.4 |
| **+ Ours** | **83.8** | **69.9** | **64.5** | **2326.6** | **84.5** | **71.3** | **85.7** | **83.8** | **83.9** | **95.1** |

Table 3: Benchmark performance comparison on visual perception tasks.

| Model | Visual Grounding Task | | | | | Vision Centric Task | | | Visual Hallucination Task | |
|---|---|---|---|---|---|---|---|---|---|---|
| | RefCOCO/+/g | LISA | RefGTA | $OD^{VG}$ | OVDEval | MMVP | CVBench | CLEVER | CHAIR↓ | POPE↑ |
| LLaVA-1.5-7B | 76.2 | 44.2 | 64.1 | 19.4 | 22.7 | 3.1 | 57.4 | 43.6 | 44.7 | 85.6 |
| +SFT | 77.1 | 44.7 | 64.6 | 20.2 | 23.0 | 9.7 | 57.8 | 44.1 | 45.2 | 85.7 |
| **+ Ours** | **78.9** | **50.1** | **66.2** | **24.8** | **27.1** | **15.1** | **60.4** | **46.6** | **41.7** | **86.4** |
| Qwen2.5-VL-3B | 84.2 | 55.3 | 70.8 | 32.1 | 39.5 | 50.4 | 67.3 | 68.7 | 35.6 | 86.1 |
| +SFT | 84.6 | 55.3 | 71.0 | 32.5 | 39.9 | 52.1 | 68.1 | 70.0 | 35.4 | 86.4 |
| **+ Ours** | **86.8** | **58.1** | **72.9** | **36.7** | **43.8** | **54.9** | **70.1** | **72.5** | **32.6** | **87.4** |
| InternVL2-2B | 77.8 | 46.1 | 66.4 | 21.7 | 24.9 | 39.6 | 56.5 | 57.1 | 37.8 | 86.2 |
| +SFT | 78.1 | 45.6 | 66.9 | 23.2 | 25.3 | 40.4 | 57.2 | 57.9 | 37.9 | 86.0 |
| **+ Ours** | **80.1** | **48.2** | **68.9** | **26.6** | **29.9** | **42.7** | **59.2** | **59.6** | **34.3** | **87.1** |
| Qwen2.5VL-7B | 87.1 | 60.3 | 74.4 | 39.3 | 44.8 | 55.1 | 73.6 | 74.4 | 32.6 | 88.9 |
| +SFT | 87.3 | 60.1 | 74.8 | 39.7 | 44.4 | 55.8 | 73.8 | 74.8 | 33.1 | 89.2 |
| **+ Ours** | **88.2** | **63.6** | **76.0** | **42.1** | **47.2** | **57.0** | **75.2** | **75.9** | **29.7** | **89.6** |
| Qwen2.5VL-32B | 89.8 | 65.9 | 77.5 | 43.1 | 49.3 | 60.4 | 77.2 | 78.5 | 28.2 | 90.3 |
| +SFT | 89.7 | 63.3 | 77.8 | 43.3 | 49.3 | 60.8 | 77.1 | 78.7 | 30.1 | 90.1 |
| **+ Ours** | **90.6** | **67.4** | **79.3** | **44.5** | **51.8** | **62.5** | **79.0** | **80.0** | **21.2** | **90.6** |

benchmarks such as LISA and OVDEval, DocVQA, CVBench, and CLEVER, SFT yields marginal improvements, whereas RAR achieves substantial gains, highlighting its superiority in bolstering visual capabilities. Furthermore, on hallucination tasks, SFT results occasionally fall below those of the base models, which we attribute to iterative training across diverse tasks exacerbating the model's bias toward linguistic priors. RAR effectively circumvents this issue, ensuring reliable visual comprehension. Additionally, Table 4 illustrates that SADS significantly enhances inference speed compared to the base models and SFT, attributable to the reduction in computations associated with sink heads' attention.

## 6.3 ABLATION STUDIES

To directly assess whether the RAR resolves the suboptimal output layer issue, we compare the layer-wise performance of the base model, SFT, and RAR on the OVDEval test set, as shown in Figure 8c. Evidently, unlike the performance degradation observed in later layers for both the base model and SFT, RAR achieves higher performance and demonstrate progressive capability improvements across all later layers. As depicted in Figure 8a, we contrast the visual attention maps in later layers, revealing that RAR effectively directs visual attention toward semantically relevant regions, in stark contrast to the persistent

Table 4: Comparative analysis of inference efficiency and accuracy.

| Method | Latency↓ | Accuracy↑ |
|---|---|---|
| Qwen2.5-VL-3B | 1.332 | 39.5 |
| + SFT | 1.332 | 39.9 |
| **+ Ours** | **1.195** | **43.8** |

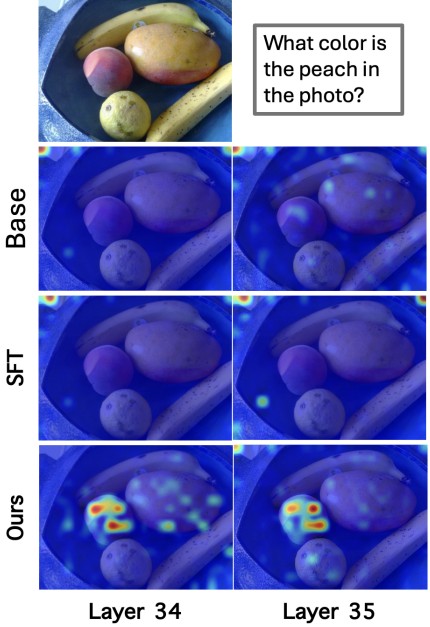
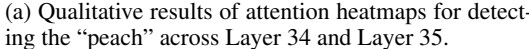

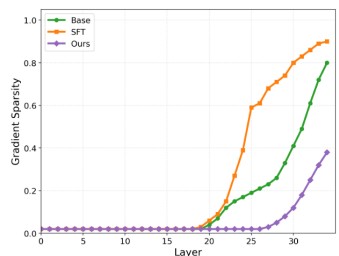

(b) Layer-wise Attention Gradient Sparsity: base model, SFT Model and Our Method.

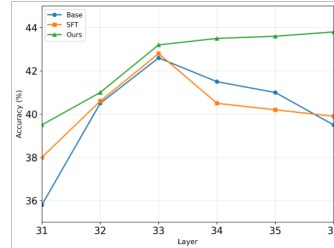

(c) Performance comparison across different layers on the OVDEval benchmark: Base model, SFT, and our method.

(a) Qualitative results of attention heatmaps for detecting the "peach" across Layer 34 and Layer 35.

Figure 8: Comprehensive ablation studies. (a) Qualitative attention heatmaps. (b) Attention gradient sparsity comparison. (c) Performance comparison across different layers.

visual attention re-sinking in the base model and SFT. In Figure 8b, we illustrate the cross-layer variations in gradient sparsity, demonstrating that RAR mitigates gradient sparsity issues in later layers, whereas the base model and SFT suffer from severe sparsity therein. Furthermore, SFT exhibits even more pronounced attention gradient sparsity compared to the base model, substantiating that training iterations exacerbate attention gradient sparsity.

We are particularly interested in assessing the scalability of SADS with respect to training data volume. To this end, we perform a comprehensive analysis of its performance across varying dataset sizes. As depicted in Figure 9, experiments on the OVDEval benchmark show that as training data increases, SFT yields diminishing returns, with performance gains plateauing markedly. In contrast, SADS demonstrates a steeper performance ascent, underscoring the visual attention re-sinking phenomenon severely impedes effective data scaling in existing MLLMs, while SADS effectively mitigates this issue and unlocks superior scaling potential.

# 7 CONCLUSION

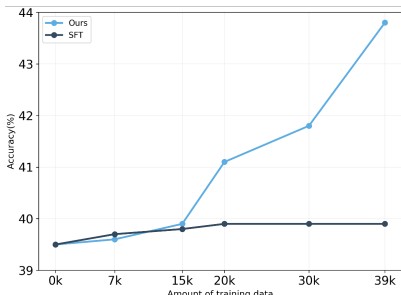

Figure 9: Influence of different head selection strategies on model performance.

In this work, we have elucidated the root causes of suboptimal output layers in MLLMs, ascribing them to attention gradient sparsity precipitated by textual supervision dominance during training. This sparsity engenders the ***Visual Attention Re-sinking*** phenomenon, head bifurcation, and a progressive disregard for visual cues, ultimately culminating in degraded output performance. To mitigate these issues, we introduce the parameter-free SADS framework, which dynamically retains all vision heads while sparsifying sink heads and ensuring model stability through a designated shared head. Comprehensive experiments spanning 20 benchmarks across five diverse task categories demonstrate that SADS surpasses standard supervised fine-tuning in performance while accelerating inference by 10.3%.

## ACKNOWLEDGEMENTS

This work was partly supported by the Special Foundations for the Development of Strategic Emerging Industries of Shenzhen(No.KJZD20231023094700001).

## ETHICS STATEMENT

Our work is in accordance with the ICLR Code of Ethics. This research did not involve human participants or animal testing. The datasets employed in this work are all publicly available and were utilized in accordance with their original licensing and usage terms. These include the training datasets (RefCOCO, Dcube, VG, GQA, OCR-VQA, Text-VQA, CLEVER) and the evaluation benchmarks (RefCOCO/g/+, LISA, RefGTA, COCO, OVDEval, VQA_v2, VizWiz, VQA_vg, GQA, MME, MMB, MMStar, AI2D, InfoVQA, TextVQA, DocVQA, MMVP, CVBench, CLEVER, POPE, CHAIR). Our methodology was designed to mitigate potential biases and avoid discriminatory results. The data used contains no personally identifiable information, and our experiments do not pose any privacy or security risks. We uphold the principles of research integrity and transparency in our work.

## REPRODUCIBILITY STATEMENT

Details regarding the setup, such as model hyperparameters and training procedures, are outlined in the main paper and appendix to ensure reproducibility. A comprehensive description of our SADS framework, including the specific layer selection criteria for each base model, is also included to aid in replication. All evaluation benchmarks used across our five task categories—visual grounding, general VQA, OCR-related VQA, vision-centric, and visual hallucination—are established public datasets, namely RefCOCO/g/+, LISA, RefGTA, COCO, OVDEval, VQA_v2, VizWiz, VQA_vg, GQA, MME, MMB, MMStar, AI2D, InfoVQA, TextVQA, DocVQA, MMVP, CVBench, CLEVER, POPE, and CHAIR, allowing for consistent re-evaluation.

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

## A  LLM USAGE

A Large Language Model (LLM) was employed for language refinement and editing of this manuscript. Its role was to improve clarity, refine phrasing, and check for grammatical consistency across the paper. The conceptual and experimental aspects of this research, including the core ideas, methodology, and data analysis, were conducted exclusively by the human authors. The LLM's assistance was confined to improving the manuscript's linguistic quality and was not used for generating scientific content. The authors retain full responsibility for all content, including any text assisted by the LLM, and have verified its scientific accuracy and originality.

## B  ADDITIONAL ABLATION STUDIES

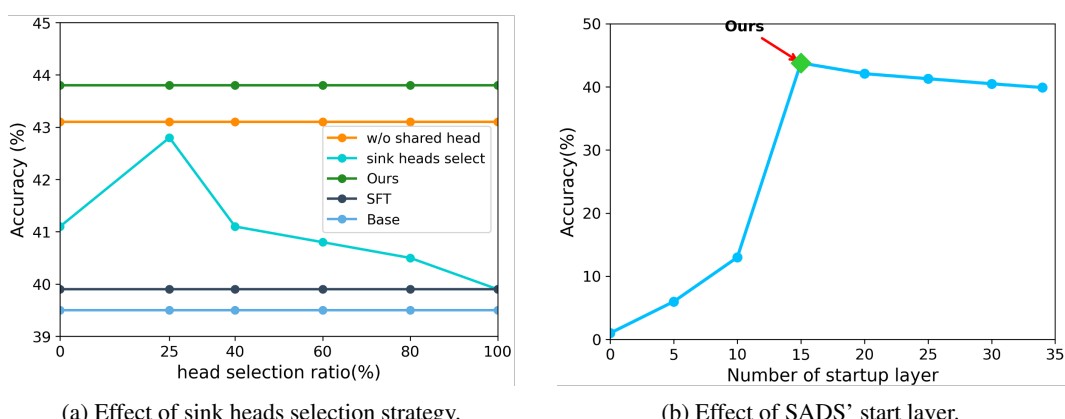

(a) Effect of sink heads selection strategy.

(b) Effect of SADS' start layer.

Figure 10: **Comprehensive ablation studies on OVDEval benchmark.** (a) Effect of Head Selection Strategy. (b) Effect of SADS' Start Layer.

### B.1  ABLATION STUDIES ON HEAD SELECTION STRATEGY AND START LAYER.

In the preceding experiments, the activation layer for SADS is dynamically determined by the onset of variance fluctuations in attention keys ($\mathbf{K}$) and gradient sparsity, while head selection leverages the valley of a bimodal Gaussian distribution fitted to maximum visual attention values (for distinguishing vision and sink heads) and non-vision token cross-attention entropy (for subdividing sink heads into $sink_G$ and $sink_S$). To evaluate the effectiveness and robustness of these adaptive mechanisms, we perform ablation studies employing fixed thresholds for both the number of activated layers and sink head retention ratios.

Table 5: Comprehensive ablation studies on head selection strategy.

| Method | OVDEval↑ | RefCOCO↑ | GQA↑ |
|---|---|---|---|
| Base | 39.5 | 84.2 | 60.4 |
| + SFT | 39.9 | 84.6 | 62.0 |
| + 0% sink heads | 41.1 | 85.0 | 62.8 |
| + 40% sink heads | 41.1 | 85.8 | 63.5 |
| + 80% sink heads | 40.8 | 85.1 | 62.2 |
| + 100% sink heads | 39.9 | 84.6 | 62.0 |
| Ours(fixed 25% sink heads) | 42.8 | 86.1 | 63.5 |
| **Ours** | **43.8** | **86.8** | **64.2** |

As shown in Figure 10a, we first assess the impact of varying sink head retention ratios on performance, revealing that our dynamic selection method, based on non-vision token cross-attention entropy, achieves markedly superior results. Among fixed-ratio strategies, retaining approximately 25% of sink heads yields optimal outcomes, consistent with the Pareto principle, whereas higher or lower ratios lead to performance degradation. Additionally, omitting the designation of a shared head (w/o shared) results in consistent declines across benchmarks, underscoring the critical role of a fixed-position shared

head in maintaining model stability. To further validate these findings across diverse benchmarks, Table 5 presents comprehensive ablation studies on head selection strategies, demonstrating that our method (Ours) outperforms all fixed-ratio variants and the base model. These ablations collectively validate the superiority of SADS's adaptive design.

As depicted in Figure 10b, ablating the SADS activation layer further demonstrates that starting from a lower layer (thus activating more layers) induces substantial degradation and garbled outputs, arising from unnecessary sparsification in well-aligned early layers. Conversely, starting from a higher layer (activating fewer layers) causes performance drops due to unaddressed visual attention re-sinking in later layers. These findings highlight the advantages of our dynamic layer selection in balancing modality alignment preservation with targeted late-layer optimization, with the optimal startup layer corresponding to the performance peak observed in the ablation curve.

Table 6: Ablation study across different iterations.

| Method | OVDEval↑ | RefCOCO↑ | GQA↑ |
|---|---|---|---|
| Base | 39.5 | 84.2 | 60.4 |
| SFT (14000 iters) | 39.9 | 84.6 | 62.0 |
| **Ours (14000 iters)** | **43.8** | **86.8** | **64.2** |
| SFT (13000 iters) | 39.9 | 84.5 | 61.5 |
| **Ours (13000 iters)** | **43.4** | **86.2** | **63.8** |
| SFT (12000 iters) | 39.6 | 84.3 | 61.7 |
| **Ours (12000 iters)** | **43.4** | **85.7** | **63.2** |
| SFT (11000 iters) | 39.6 | 84.5 | 61.0 |
| **Ours (11000 iters)** | **42.8** | **85.8** | **63.3** |
| SFT (10000 iters) | 39.7 | 84.4 | 60.8 |
| **Ours (10000 iters)** | **42.3** | **85.6** | **62.5** |

### B.2 ABLATION STUDY ON TRAINING STEPS.

As shown in Table 6, to investigate whether earlier checkpoints during training yield superior performance, we evaluate and compare model performance across varying training iterations. Key observations include: (1) Earlier checkpoints exhibit performance degradation, as prematurely halting training mitigates gradient sparsity issues but exacerbates under-training, leading to suboptimal results; (2) Across all checkpoints, our method consistently outperforms SFT, underscoring its robust performance and stability.

Table 7: Ablation study across different training objectives.

| Method | OVDEval↑ | RefCOCO↑ | GQA↑ |
|---|---|---|---|
| Base | 39.5 | 84.2 | 60.4 |
| SFT | 39.9 | 84.6 | 62.0 |
| w/ regularization | 40.8 | 85.3 | 62.7 |
| w/ attention reweight | 40.6 | 85.0 | 62.3 |
| **Ours** | **43.8** | **86.8** | **64.2** |

### B.3 ABLATION STUDY ON DIFFERENT TRAINING OBJECTIVES.

As shown in Table 7, we further examine the performance under different training objectives. We explore two alternative strategies: (1) **regularization**; regularization to elevate the weights of vision heads, implemented via a KL divergence prior that encourages vision heads to align with a high-weight distribution ($\mathcal{N}(1.2, 0.1)$) while constraining sink heads to a low-weight distribution ($\mathcal{N}(0.3, 0.1)$), formally expressed as $\mathcal{L}_{reg} = \sum_{h \in \text{vision}} \text{KL}(\alpha_h || \mathcal{N}(\mu_v, \sigma_v^2)) + \sum_{h \in \text{sink}} \text{KL}(\alpha_h || \mathcal{N}(\mu_s, \sigma_s^2))$; (2) **attention reweight**; attention reweighting on sink tokens within sink heads to prioritize semantically relevant visual information, achieved by dynamically scaling

attention weights based on visual saliency metrics, reweighting as $a'_{i,j} = a_{i,j} \cdot s_j / \sum_k a_{i,k} \cdot s_k$, where $s_j$ denotes the saliency score of token $j$ (upweighting edges toward high-semantic regions while downweighting low-information backgrounds). The results indicate that both approaches yield modest improvements over vanilla SFT. However, our SADS framework substantially outperforms these variants. This superiority stems from the root cause of suboptimal output layers: the Visual Attention Re-sinking phenomenon within sink heads, which impairs model performance by marginalizing visual cues. While regularization and reweighting provide temporary alleviation of sinking, the issue inevitably re-emerges with deeper training iterations. In contrast, our sparsification approach fundamentally eradicates re-sinking, thereby maximizing the model's potential.

### B.4 STATISTICAL ANALYSIS IN SINK HEADS.

As illustrated in the Figure 11, we randomly sample 2,000 heads and compute their non-image token cross-attention entropy, revealing a pronounced bimodal Gaussian distribution characterized by distinct low-entropy ($sink_S$ heads) and high-entropy ($sink_G$ heads) peaks. Consequently, for each layer, we model the entropy distribution across all sink heads using a GMM and employ the valley as a dynamic threshold for differentiation.

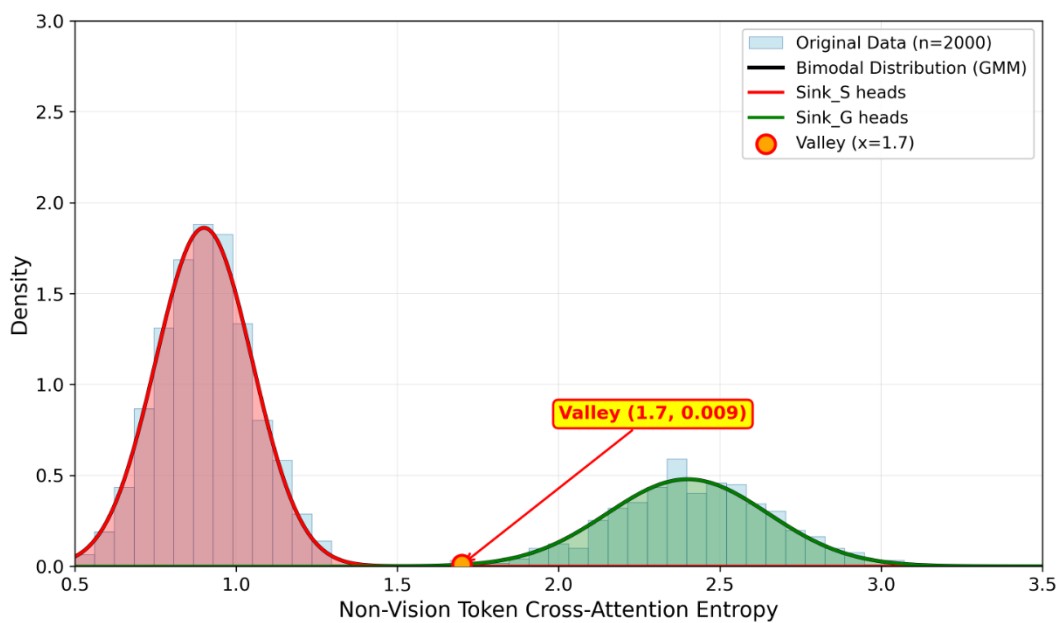

Figure 11: Distribution of non-vision token cross-attention entropy for $Sink_G$ heads and $Sink_S$ heads, showing a bimodal distribution.

## C LIMITATIONS AND FUTURE WORK

Although our method effectively maximizes model capacity at current parameter scales, we acknowledge its inherent limitations. While sparsifying redundant gradient spaces has proven efficacious, infilling techniques may offer a more robust alternative. In future work, we plan to integrate the SADS framework into a unified generation-understanding paradigm, leveraging generative capabilities to populate sparse spaces and further enhance multimodal fusion.

## D ADDITIONAL QUALITATIVE RESULTS

We provide additional Qualitative results of cross layer visual attention map in the Figure 12 and low-entropy $sink_S$ and high-entropy $sink_G$ in Figure 13.

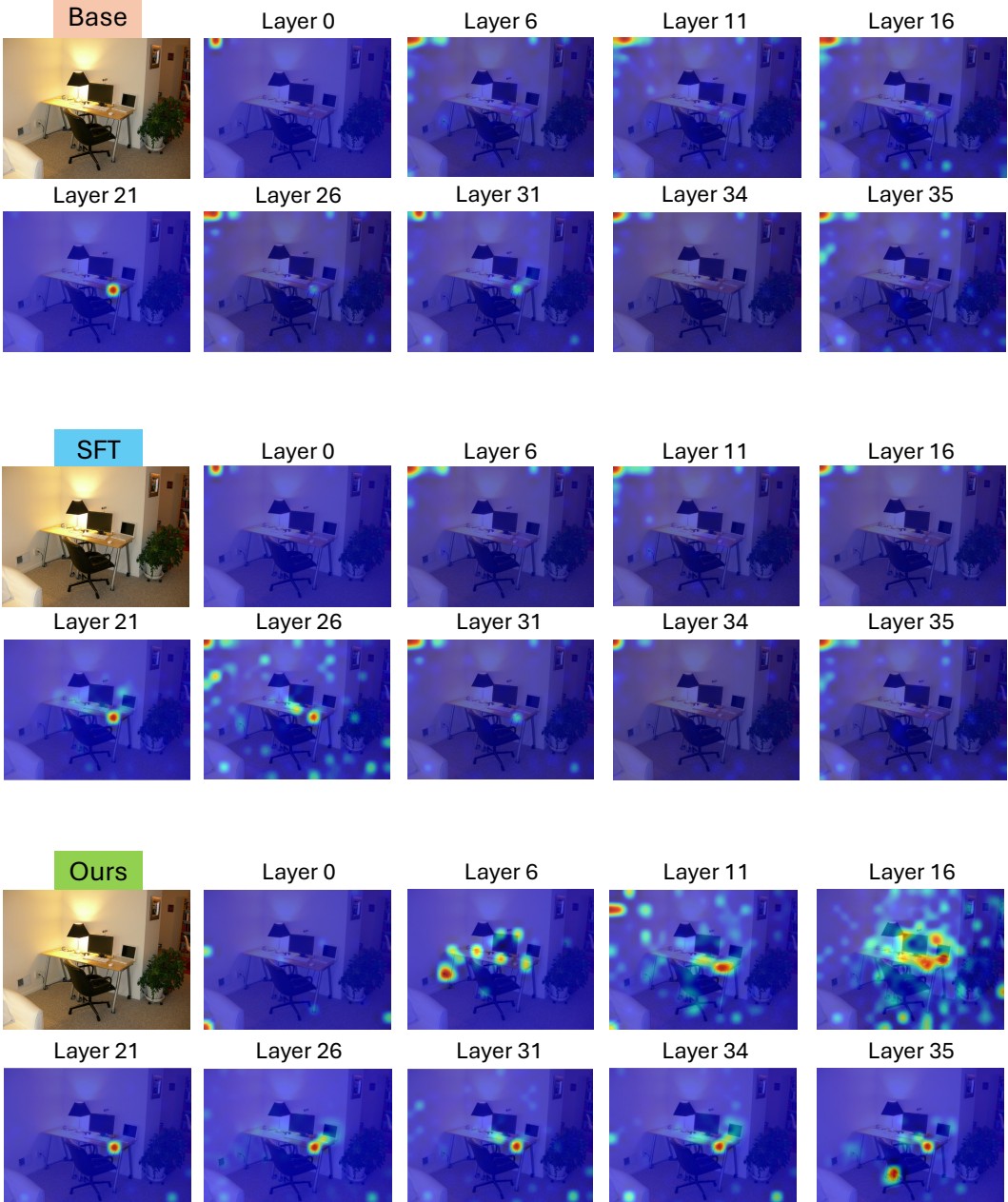

Figure 12: Qualitative results of visual attention heatmaps for detecting the "mouse" across Layers.

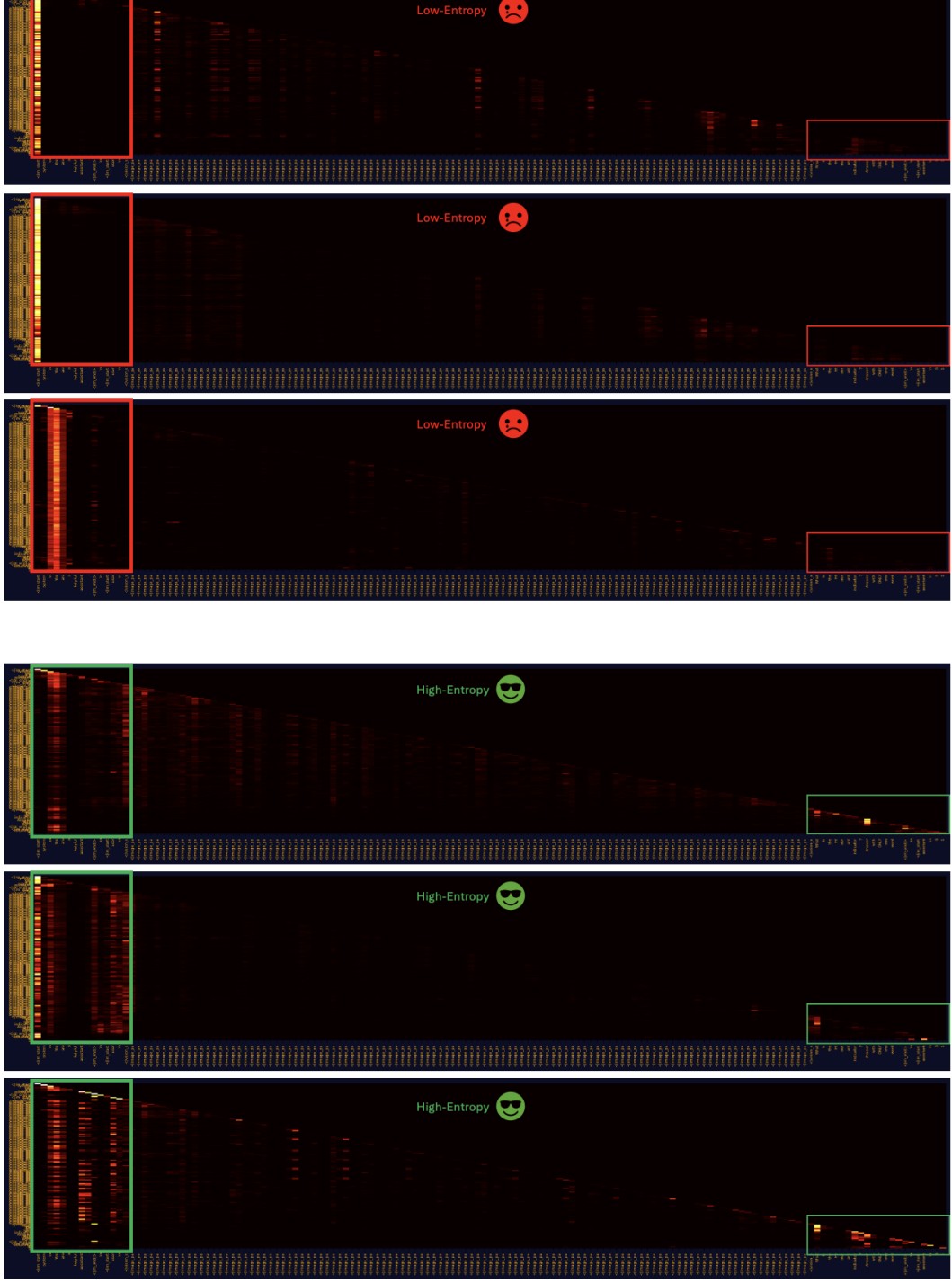

Figure 13: Qualitative results of low-entropy $sink_S$ and high-entropy $sink_G$

