# OpenReview forum: "RAR: Reversing Visual Attention Re-Sinking for  Unlocking Potential in Multimodal Large Language Models"
_ICLR.cc/2026/Conference — ICLR 2026 Poster_

### Official Review · Reviewer_VNAx · 2025-10-22

**Soundness:** 3
**Presentation:** 3
**Contribution:** 3
**Rating:** 4
**Confidence:** 4

**Summary:**

The paper presents RAR: a parameter-free method to address the problem of re-sinks in the final layers of a vision-language model, that cause the model to underperform due to undesired attentions. Unhealthy attention heads are removed, which improves the performance.

**Strengths:**

The observations about the problems in the final layers, the degradation of performance because of that, the illustrations of spurious visual attention, are all interesting. The finding that visual attention is bi-modal and separates the vision heads and sink heads, is interesting too. The performance gains are structural but marginal. The proposed method is effective and simple, which makes the approach feasible for a broader audience.

**Weaknesses:**

The illustrations are sparse (e.g. Figures 3c, 4a and 8a) and sufficient to convey the main idea, but they are redundant and simplistic, as a consequence a better understanding of the separation into visual and attention heads is not provided. More importantly, the distinction between useful and unhealthy sink heads is not illustrated, and details are lacking to understand this better.

The whole idea of the paper is to remove the relatively small subset of those unhealthy sink heads. A better understanding of that subset, which ones they are, why they hurt the performance; this is fundamental for a paper that builds on those principles.

**Questions:**

Which subset of the attention heads is unhealthy and why is that, why does that hurt the performance of the last layers? Can you provide insights with some illustrations, and with some statistics, e.g. are they mostly on particular areas in the image, or semantic regions?

---

> ### Author Response · Authors · 2025-11-26
> **Response to Reviewer VNAx (Part 1/2)**
>
> Thank you for your insightful suggestions and positive evaluation of our work. Below, we address your concerns point-by-point.
>
> ---
>
> >**W1: “The illustrations are sparse (e.g. Figures 3c, 4a and 8a) and sufficient to convey the main idea, but they are redundant and simplistic, as a consequence a better understanding of the separation into visual and attention heads is not provided. More importantly, the distinction between useful and unhealthy sink heads is not illustrated, and details are lacking to understand this better.”**
>
> **Response:** Thank you for your detailed review and valuable feedback. Following your suggestions, we have refined the figures and associated descriptions, including **Figures 3c, 4a, and 8a**, to better highlight their respective roles and enhance clarity.
>
> Specifically:
>
> (1) Each heatmap in **Figure 3c** is derived by averaging the attention maps across all heads in that layer, illustrating the processes of visual attention sinking, sink diminishing, and re-sinking across layers. To emphasize whether visual attention is sinking in each layer, we have added green boxes to annotate semantically relevant regions.
>
> (2) We have updated **Figure 4a**, per your recommendation, to showcase that, within the same layer, there exist distinct patterns among attention heads: vision heads focus visual attention on semantically relevant areas, while sink heads concentrate on semantically irrelevant background regions. Additionally, we have revised **Figure 4b** to better link with **Figure 4a** and illustrate the separation between vision heads and sink heads.
>
> (3) Each heatmap in **Figure 8a** is obtained by averaging the attention maps across all heads in that layer. This visualization demonstrates that our method successfully directs the model's visual attention toward semantically relevant regions, whereas the base model and SFT approaches continue to emphasize semantically irrelevant backgrounds.

---

> > ### Author Response · Authors · 2025-11-26
> > **Response to Reviewer VNAx (Part 2/2)**
> >
> > >**W2&Q: “...A better understanding of that subset, which ones they are, why they hurt the performance...Which subset of the attention heads is unhealthy and why is that, why does that hurt the performance of the last layers? Can you provide insights with some illustrations, and with some statistics...”**
> >
> > **Response:** Thank you for your detailed and insightful suggestions. Inspired by your feedback, we conducted an in-depth analysis of patterns within sink head subsets, distinguishing between $sink_G$ heads (which preserve global and contextual information) and $sink_S$ heads (which sink attention onto individual low-semantic tokens). By dynamically removing $sink_S$ heads using an adaptive threshold, we achieved superior performance, as detailed in the red-highlighted sections of **Sections 4.2 and 5**, along with detailed statistics, visualizations, and case studies in **Appendices B.4 and D**.
> >
> > To provide a clearer understanding, we categorize attention heads hierarchically: at the top level, heads are divided into vision heads and sink heads; within sink heads, they are further subdivided into $sink_G$ (retained for their benefits) and $sink_S$ (discarded due to their harm). Specifically:
> >
> > (1) **Vision heads:** Focus on semantically relevant visual regions, providing essential visual perception.
> >
> > (2) **$Sink_G$ heads:** Exhibit uniformly dispersed attention across non-vision tokens, preserving global and contextual information.
> >
> > (3) **$Sink_S$ heads:** Concentrate attention on individual low-semantic tokens, disrupting modality fusion.
> >
> > For sink heads in particular, we compute cross-attention among all tokens and observe distinct patterns in non-vision token cross-attention entropy. $Sink_S$ heads disrupt effective modality fusion in the final layers by marginalizing visual cues and biasing outputs toward textual priors, ultimately degrading performance. We quantify this via the entropy of the attention distribution, defined as follows. Let $\mathbf{A} \in \mathbb{R}^{L_q \times L_k}$ denote the attention matrix for a given head, where $L_q$ and $L_k$ are the query and key sequence lengths, respectively. Let $\mathcal{I}$ be the set of indices corresponding to non-vision tokens (keys). Extract the submatrix $\mathbf{A} _ \text{sub} = \mathbf{A}[:, \mathcal{I}] \in \mathbb{R}^{L _ q \times |\mathcal{I}|}$. Renormalize each row $i$ such that
> > $$\mathbf{A} _ \text{sub}[i, :] \leftarrow \frac{\mathbf{A} _ \text{sub}[i, :]}{\sum _ {j \in \mathcal{I}} \mathbf{A} _ \text{sub}[i, j]},$$
> > ensuring each row sums to 1 over non-image keys. Then, compute the average attention distribution $\mathbf{p} \in \mathbb{R}^{|\mathcal{I}|}$ where
> > $$
> > p _ j = \frac{1}{L _ q} \sum_{i=1}^{L _ q} \mathbf{A} _ \text{sub}[i, j]
> > $$
> > for each $j \in \mathcal{I}$. The entropy is given by
> > $$
> > H = -\sum_{j \in \mathcal{I}} p_j \log p_j.
> > $$
> >
> > We define high-entropy heads as $sink_G$ and low-entropy heads as $sink_S$. As shown in the table below on the OVDEval benchmark, adding an extra $sink_S$ head or ablating vision/$sink_G$ heads leads to significant drops, underscoring the harm of isolated token sinking and the value of visual cues/global context:
> >
> > | **Method**               | **Accuracy (%)** |
> > |--------------------------|------------------|
> > | Qwen2.5-VL-3B            | 39.5             |
> > | w/o $sink_S$ head        | 43.8             |
> > | w/ 1 $sink_S$ head       | 43.0             |
> > | w/o 1 $sink_G$ head      | 43.2             |
> > | w/o 1 vision head        | 42.6             |
> >
> > Moreover, entropy values across heads follow a bimodal Gaussian distribution (visualized in **Appendix B.4**), allowing us to leverage its valley as a dynamic threshold for differentiating $sink_S$ and $sink_G$ heads. This adaptive approach further boosts performance by precisely retaining beneficial heads while eliminating harmful ones.
> >
> > ---
> >
> > We sincerely thank you for your valuable suggestions, which have helped us refine and strengthen the paper. If you have any additional questions or concerns, please do not hesitate to let us know.

---

> > > ### Comment · Reviewer_VNAx · 2025-11-26
> > > **reply to rebuttal**
> > >
> > > I am satisfied with the detailed explanations, clarifications, and paper modifications. I will increase my score.

---

> > > > ### Author Response · Authors · 2025-11-27
> > > > **Thanks for your timely and valuable reply!**
> > > >
> > > > Thank you for your valuable and timely response! We are delighted that our rebuttal has addressed your concerns and appreciate your positive acknowledgment. We sincerely thank you for the effort and time you invested during the review and discussion phases, this exchange has been immensely beneficial to us. Thank you once again!

---

### Official Review · Reviewer_7PSW · 2025-10-30

**Soundness:** 3
**Presentation:** 3
**Contribution:** 3
**Rating:** 4
**Confidence:** 4

**Summary:**

This paper focus on the problem that the intermediate decoder layers outperform the final ones in MLLMs, and attribute this issue to the Visual Attention Re-sinking phenomenon. Then, a parameter-free Sink Attention Dynamic Sparsification framework is proposed. The proposed SADS achieves superior effectiveness and inference efficiency on several benchmarks.

**Strengths:**

1. It deeply identifies text-only supervision as the cause of suboptimal MLLM output layers.
2. The SADS framework effectively addresses key issues to optimize output layers.
3. Comprehensive experiments across 20 benchmarks validate its superiority.

**Weaknesses:**

1. From Figure 2 and Table 1, it can be observed that the performance degradation caused by the re-sinking phenomenon is limited. It can also be seen from the experiments that the improvement brought by the proposed method is limited.
2. To demonstrate its effectiveness and  generalizability, more models of different sizes (7B, 13B, ...) should be tested.
3. An ablation experiment on the proportion of sink heads should be added; additionally, is it reasonable to use a fixed proportion?

**Questions:**

Refer to Weaknesses.

---

> ### Author Response · Authors · 2025-11-26
> **Response to Reviewer 7PSW (Part 1/3)**
>
> Thank you for your valuable insights and positive evaluation of our work. Below, we address your concerns point-by-point.
>
> ---
>
> >**W1: “From Figure 2 and Table 1, it can be observed that the performance degradation caused by the re-sinking phenomenon is limited. It can also be seen from the experiments that the improvement brought by the proposed method is limited.”**
>
> **Response:** Thank you for your insightful comment. To more precisely quantify the impact of the visual attention re-sinking phenomenon, we revisited the experiments in **Figure 2**. We also incorporated the expected output layer performance in the absence of visual attention re-sinking, which verifies the extent of the performance limitations imposed by this issue. Specifically, we applied targeted fine-tuning to the output heads across all layers, aligning their predictions with ground-truth annotations. This controlled setup isolates the re-sinking effects by ensuring that upstream components are optimized, revealing the substantial performance degradation induced by re-sinking across diverse baselines, creating a significant gap from expected outputs.
>
> To illustrate the severity of these limitations and the benefits of our method in overcoming them, we present the comprehensive benchmark results in **Tables 2 and 3** below:
>
> | Model                | VQA^v2 | GQA  | VQA^vg | MME    | MMB  | MMStar | AI2D | InfoVQA | TextVQA | DocVQA |
> |----------------------|------------------|------|------------------|--------|------|--------|------|----------|----------|--------|
> | LLaVA-1.5-7B         | 78.3             | 61.1 | 54.6             | 1808.4 | 61.1 | 33.2   | 55.7 | 41.2     | 64.7     | 69.4   |
> | +SFT                 | 79.1             | 63.1 | 55.7             | 1899.6 | 61.9 | 34.5   | 56.2 | 43.7     | 65.5     | 71.2   |
> | **+ Ours**           | **80.8**         | **65.2** | **58.5**     | **2018.8** | **63.2** | **36.3** | **57.1** | **46.3** | **67.5** | **74.4** |
> | Qwen2.5VL-3B         | 76.7             | 60.4 | 54.3             | 2184.1 | 75.4 | 53.0   | 77.9 | 75.1     | 78.7     | 93.0   |
> | +SFT                 | 77.9             | 62.0 | 55.2             | 2199.9 | 75.9 | 53.7   | 78.4 | 75.9     | 79.0     | 92.9   |
> | **+ Ours**           | **79.7**         | **64.2** | **58.1**     | **2276.3** | **76.9** | **55.4** | **79.5** | **77.3** | **80.4** | **93.5** |
> | InternVL2-2B         | 72.9             | 55.6 | 50.1             | 1864.3 | 69.1 | 48.9   | 73.1 | 58.8     | 73.4     | 86.4   |
> | +SFT                 | 74.2             | 56.9 | 52.3             | 1899.1 | 70.0 | 49.6   | 73.9 | 59.1     | 73.8     | 86.6   |
> | **+ Ours**           | **75.9**         | **59.0** | **55.4**     | **2006.5** | **71.6** | **50.8** | **75.7** | **60.5** | **75.9** | **88.2** |
> | Qwen2.5VL-7B         | 81.6             | 65.8 | 60.5             | 2276.3 | 82.2 | 64.2   | 84.1 | 81.7     | 80.2     | 94.8   |
> | +SFT                 | 81.9             | 66.1 | 61.0             | 2230.2 | 82.0 | 64.5   | 84.4 | 82.0     | 80.7     | 94.2   |
> | **+ Ours**           | **82.6**         | **67.9** | **62.1**     | **2289.8** | **83.3** | **66.0** | **84.8** | **82.9** | **81.3** | **95.0** |
> | Qwen2.5VL-32B        | 82.9             | 68.4 | 63.6             | 2297.4 | 83.8 | 70.3   | 85.2 | 83.4     | 82.8     | 94.8   |
> | +SFT                 | 83.0             | 68.6 | 63.9             | 2255.4 | 83.8 | 69.6   | 85.1 | 83.0     | 82.9     | 94.4   |
> | **+ Ours**           | **83.8**         | **69.9** | **64.5**     | **2326.6** | **84.5** | **71.3** | **85.7** | **83.8** | **83.9** | **95.1** |

---

> ### Author Response · Authors · 2025-11-26
> **Response to Reviewer 7PSW (Part 2/3)**
>
> | Model                | RefCOCO/+/g | LISA | RefGTA | OD^VG | OVDEval | MMVP | CVBench | CLEVER | CHAIR↓ | POPE↑ |
> |----------------------|-------------|------|--------|------------------|----------|------|----------|--------|-------------------|------------------|
> | LLaVA-1.5-7B         | 76.2        | 44.2 | 64.1   | 19.4             | 22.7     | 3.1  | 57.4     | 43.6   | 44.7              | 85.6             |
> | +SFT                 | 77.1        | 44.7 | 64.6   | 20.2             | 23.0     | 9.7  | 57.8     | 44.1   | 45.2              | 85.7             |
> | **+ Ours**           | **78.9**    | **50.1** | **66.2** | **24.8**       | **27.1** | **15.1** | **60.4** | **46.6** | **41.7**        | **86.4**         |
> | Qwen2.5-VL-3B        | 84.2        | 55.3 | 70.8   | 32.1             | 39.5     | 50.4 | 67.3     | 68.7   | 35.6              | 86.1             |
> | +SFT                 | 84.6        | 55.3 | 71.0   | 32.5             | 39.9     | 52.1 | 68.1     | 70.0   | 35.4              | 86.4             |
> | **+ Ours**           | **86.8**    | **58.1** | **72.9** | **36.7**       | **43.8** | **54.9** | **70.1** | **72.5** | **32.6**        | **87.4**         |
> | InternVL2-2B         | 77.8        | 46.1 | 66.4   | 21.7             | 24.9     | 39.6 | 56.5     | 57.1   | 37.8              | 86.2             |
> | +SFT                 | 78.1        | 45.6 | 66.9   | 23.2             | 25.3     | 40.4 | 57.2     | 57.9   | 37.9              | 86.0             |
> | **+ Ours**           | **80.1**    | **48.2** | **68.9** | **26.6**       | **29.9** | **42.7** | **59.2** | **59.6** | **34.3**        | **87.1**         |
> | Qwen2.5VL-7B         | 87.1        | 60.3 | 74.4   | 39.3             | 44.8     | 55.1 | 73.6     | 74.4   | 32.6              | 88.9             |
> | +SFT                 | 87.3        | 60.1 | 74.8   | 39.7             | 44.4     | 55.8 | 73.8     | 74.8   | 33.1              | 89.2             |
> | **+ Ours**           | **88.2**    | **63.6** | **76.0** | **42.1**       | **47.2** | **57.0** | **75.2** | **75.9** | **29.7**        | **89.6**         |
> | Qwen2.5VL-32B        | 89.8        | 65.9 | 77.5   | 43.1             | 49.3     | 60.4 | 77.2     | 78.5   | 28.2              | 90.3             |
> | +SFT                 | 89.7        | 63.3 | 77.8   | 43.3             | 49.3     | 60.8 | 77.1     | 78.7   | 30.1              | 90.1             |
> | **+ Ours**           | **90.6**    | **67.4** | **79.3** | **44.5**       | **51.8** | **62.5** | **79.0** | **80.0** | **21.2**        | **90.6**         |
>
> These results show that our method achieves substantial performance gains across diverse baselines, particularly on vision-centric tasks. For instance, using LLaVA-1.5-7B as the baseline, our method improves LISA, OD, OVDEval, CVBench, CHAIRᵢ, and MMB by **+5.9, +5.4, +4.4, +3.0, -3.0 (lower is better for CHAIRᵢ), and +2.1**, respectively, while SFT yields minimal or even negative changes **(+0.5, +0.8, +0.3, +0.4, +0.5, +0.8)**. Similarly, with Qwen2.5VL-3B, our method boosts these metrics by **+2.8, +4.6, +4.3, +2.8, -3.0, and +1.5**, compared to SFT's modest gains **(0.0, +0.4, +0.4, +0.8, -0.2, +0.5)**. Additionally, as analyzed in **Figure 9**, SFT's benefits diminish with increasing training data, leading to plateaued gains. In contrast, SADS delivers more pronounced improvements, highlighting how visual attention re-sinking hinders data scaling in existing multimodal models, while our method alleviates this bottleneck and unlocks greater scaling potential.
>
> ---
>
> >**W2: “To demonstrate its effectiveness and generalizability, more models of different sizes (7B, 13B, ...) should be tested.”**
>
> **Response:** Thank you for this valuable suggestion. Following your recommendation, we extended our experiments to larger models, including Qwen2.5VL-7B and Qwen2.5VL-32B, as presented in **Tables 2 and 3** above. We observe that, when scaling to larger models, SFT yields minimal performance gains (particularly on vision-centric benchmarks), whereas our method consistently delivers substantial improvements. This demonstrates the effectiveness, stability, and scalability of our sparsification principle.

---

> ### Author Response · Authors · 2025-11-26
> **Response to Reviewer 7PSW (Part 3/3)**
>
> >**W3: “An ablation experiment on the proportion of sink heads should be added; additionally, is it reasonable to use a fixed proportion?”**
>
> **Response:** Thank you for raising this point. To address this, we conducted an ablation analysis on the fixed selection ratio for sink heads in **Appendix B.1**. Specifically, we evaluated the impact of different sink head retention strategies, with results summarized in the table below:
>
> | **Method** | **OVDEval↑** | **RefCOCO↑** | **GQA↑** |
> |------------|--------------|--------------|----------|
> | Base | 39.5 | 84.2 | 60.4 |
> | + SFT | 39.9 | 84.6 | 62.0 |
> | + 0% sink heads | 41.1 | 85.0 | 62.8 |
> | + 40% sink heads | 41.1 | 85.8 | 63.5 |
> | + 80% sink heads | 40.8 | 85.1 | 62.2 |
> | + 100% sink heads | 39.9 | 84.6 | 62.0 |
> | Ours(fixed 25% sink heads) | 42.8 | 86.1 | 63.5 |
> | **Ours** | **43.8** | **86.8** | **64.2** |
>
> We observe that retaining the top 25% of sink heads yields optimal performance across multiple benchmarks, with only mild sensitivity to the threshold.
>
> Furthermore, inspired by your suggestion, we performed an in-depth analysis of patterns within sink head subsets to refine this strategy, as detailed in the red-highlighted sections of **Sections 4.2 and 5**, along with detailed statistics, visualizations, and case studies in **Appendices B.4 and D**.
>
> We found that not all sink heads are equally beneficial: some disrupt effective modality fusion, marginalize visual cues, and bias outputs toward textual priors, ultimately degrading performance. To identify these, we compute cross-attention among all tokens within sink heads and observe distinct patterns in non-vision token cross-attention (illustrated in **Figure 5**): some heads exhibit uniformly dispersed attention across tokens (preserving global and contextual information), while others sink attention onto individual low-semantic tokens. Based on this insight, we propose a dynamic strategy that quantifies these patterns via the entropy of the non-vision token attention distribution, defined as follows.
>
> Let $\mathbf{A} \in \mathbb{R}^{L_q \times L_k}$ denote the attention matrix for a given head, where $L_q$ and $L_k$ are the query and key sequence lengths, respectively. Let $\mathcal{I}$ be the set of indices corresponding to non-vision tokens (keys). Extract the submatrix $\mathbf{A} _ \text{sub} = \mathbf{A}[:, \mathcal{I}] \in \mathbb{R}^{L _ q \times |\mathcal{I}|}$. Renormalize each row $i$ such that
> $$\mathbf{A} _ \text{sub}[i, :] \leftarrow \frac{\mathbf{A} _ \text{sub}[i, :]}{\sum _ {j \in \mathcal{I}} \mathbf{A} _ \text{sub}[i, j]},$$
> ensuring each row sums to 1 over non-image keys. Then, compute the average attention distribution $\mathbf{p} \in \mathbb{R}^{|\mathcal{I}|}$ where
> $$
> p _ j = \frac{1}{L _ q} \sum_{i=1}^{L _ q} \mathbf{A} _ \text{sub}[i, j]
> $$
> for each $j \in \mathcal{I}$. The entropy is given by
> $$
> H = -\sum_{j \in \mathcal{I}} p_j \log p_j.
> $$
>
> We define high-entropy heads as $sink_G$ heads and low-entropy heads as $sink_S$ heads. As shown in the table below on the OVDEval benchmark, adding an extra $sink_S$ head or ablating vision heads or $sink_G$ heads leads to significant performance drops, underscoring the harm of isolated token sinking and the value of visual cues and global context. Moreover, entropy values across heads follow a bimodal Gaussian distribution, enabling us to leverage its valley as a dynamic threshold for differentiating $sink_S$ and $sink_G$ heads and selectively removing the harmful ones, which further enhances our method's performance.
>
> | **Method**               | **Accuracy (%)** |
> |--------------------------|------------------|
> | Qwen2.5-VL-3B            | 39.5             |
> | w/o $sink_S$ head        | 43.8             |
> | w/ 1 $sink_S$ head       | 43.0             |
> | w/o 1 $sink_G$ head      | 43.2             |
> | w/o 1 vision head        | 42.6             |
>
> ---
>
> We sincerely thank you for your valuable suggestions, which have helped us refine and strengthen the paper. If you have any additional questions or concerns, please do not hesitate to let us know.

---

### Official Review · Reviewer_3kFx · 2025-11-01

**Soundness:** 3
**Presentation:** 3
**Contribution:** 3
**Rating:** 6
**Confidence:** 4

**Summary:**

This paper studies the root cause of the visual attention resinking phenomenon, and shows that this phenomenon is the reason why output layers of MLLM yield worse performance than middle layers, a frequently observed phenomenon in existing work. The paper finds that the visual attention resinking phenomena is caused by attention gradient sparsity, which makes the gradient distribution over attention heads sparse hence causing sink tokens to concentrate in sink heads. Based on these findings, the paper proposes a "Sink Attention Dynamic Sparsification" strategy, which selects vision heads per layer and forces model to focus on vision signals. Experimental results show improvements over baselines.

**Strengths:**

The paper developed a systematic approach to diagnose a commonly observed but not well-understood phenomenon. The analysis is logical, insightful and convincing, revealing an overlooked aspect in MLLM training and inference.

**Weaknesses:**

The proposed approach can be viewed as a patch on existing models rather than a solution to the root cause of the resinking problem. It would strengthen the paper if it also proposed ways to address the issue more fundamentally (for example, through improved training objectives).

**Questions:**

- It seems the gradient sparsity problem gets worse as the training steps increases. If this is the case, I wonder whether the MLLM performance (from the last output layer) would be better at an earlier checkpoint?
- In Fig 5a, how is the gradient sparsity precisely defined?
- The framework retains the top 25% of sink heads, is this ratio empirically optimized, and how sensitive is the model performance to this threshold?
- The tested models used are up to only 7B parameters. Would the same sparsification principle hold in larger models? Due to increased capacity of larger models, it's conceivable that the resinking problem would be alleviated.
- Would it be possible to address the gradient sparsity problem with better training objectives (e.g. adding regularization terms to upweight vision heads), which address the problem at a more fundamental level?

---

> ### Author Response · Authors · 2025-11-26
> **Response to Reviewer 3kFx (Part 1/4)**
>
> Thank you for your positive evaluation of our work and for the valuable suggestions provided. Below, we address your concerns point-by-point.
>
> ---
>
> >**W&Q5: “...It would strengthen the paper if it also proposed ways to address the issue more fundamentally...address the gradient sparsity problem with better training objectives (e.g. adding regularization terms to upweight vision heads), which address the problem at a more fundamental level?’’**
>
> **Response:** Thank you for this insightful suggestion. Inspired by your feedback, we conducted additional experiments on alternative training objectives and included quantitative analyses in **Appendix B.3**.
>
> Specifically, we explored two strategies:
>
> (1) **Regularization**: Regularization to elevate vision head weights, implemented via a KL divergence prior that encourages vision heads to align with a high-weight distribution ($\\mathcal{N}(1.2, 0.1)$) while constraining sink heads to a low-weight distribution ($\\mathcal{N}(0.3, 0.1)$), formally expressed as $\\mathcal{L}\\ _ {reg} = \\sum\\ _ {h \\in \\text{vision}} \\text{KL}(\\alpha\\ _ {h} || \\mathcal{N}(\\mu\\ _ {v}, \\sigma\\ _ {v}^{2})) + \\sum\\ _ {h \\in \\text{sink}} \\text{KL}(\\alpha\\ _ {h} || \\mathcal{N}(\\mu\\ _ {s}, \\sigma\\ _ {s}^{2}))$;
>
> (2) **Attention Reweight**: Attention reweighting on sink tokens within sink heads to prioritize semantically relevant visual information, achieved by dynamically scaling attention weights based on visual saliency metrics, reweighting as $a'\\ _ {i,j} = a\\ _ {i,j} \\cdot s\\ _ {j} / \\sum\\ _ {k} a\\ _ {i,k} \\cdot s\\ _ {k}$, where $s\\ _ {j}$ denotes the saliency score of token $j$ (upweighting edges toward high-semantic regions while downweighting low-information backgrounds).
>
> The results are summarized in the table below:
>
> | **Method** | **OVDEval↑** | **RefCOCO↑** | **GQA↑** |
> |------------|--------------|--------------|----------|
> | Base      | 39.5        | 84.2        | 60.4    |
> | SFT       | 39.9        | 84.6        | 62.0      |
> | w/ Regularization | 40.8 | 85.3 | 62.7 |
> | w/ Attention Reweight | 40.6 | 85.0 | 62.3 |
> | **Ours**  | **43.8**    | **86.8**    | **64.2** |
>
> We observe that both approaches yield modest improvements over vanilla SFT. However, our SADS framework substantially outperforms these variants. This superiority arises because the root cause of suboptimal output layers is the Visual Attention Re-sinking phenomenon within sink heads, which marginalizes visual cues and impairs performance. While regularization and reweighting offer temporary mitigation, the issue tends to re-emerge during deeper training iterations. In contrast, our sparsification approach fundamentally eliminates re-sinking, thereby unlocking the model's full potential.
>
> ---
>
> >**Q1: "...whether the MLLM performance (from the last output layer) would be better at an earlier checkpoint?"**
>
> **Response:** Thank you for this valuable suggestion. Following your advice, we added experiments to explore whether performance improves at earlier checkpoints, as detailed in **Appendix B.2**.
>
> Specifically, we compared the performance of SFT and our method across various training iteration steps. The results are summarized in the table below:
>
> | **Method** | **OVDEval↑** | **RefCOCO↑** | **GQA↑** |
> |------------|--------------|--------------|----------|
> | Base | 39.5 | 84.2 | 60.4 |
> | SFT (14000 iters) | 39.9 | 84.6 | 62.0 |
> | **Ours (14000 iters)** | **43.8** | **86.8** | **64.2** |
> | SFT (13000 iters) | 39.9 | 84.5 | 61.5 |
> | **Ours (13000 iters)** | **43.4** | **86.2** | **63.8** |
> | SFT (12000 iters) | 39.6 | 84.3 | 61.7 |
> | **Ours (12000 iters)** | **43.4** | **85.7** | **63.2** |
> | SFT (11000 iters) | 39.6 | 84.5 | 61.0 |
> | **Ours (11000 iters)** | **42.8** | **85.8** | **63.3** |
> | SFT (10000 iters) | 39.7 | 84.4 | 60.8 |
> | **Ours (10000 iters)** | **42.3** | **85.6** | **62.5** |
>
> These results indicate:
>
> (1) Earlier checkpoints exhibit performance degradation, as premature stopping alleviates gradient sparsity issues but introduces severe underfitting, leading to overall reduced performance.
>
> (2) Across all checkpoints, our method consistently outperforms SFT, demonstrating its effectiveness and stability.

---

> ### Author Response · Authors · 2025-11-26
> **Response to Reviewer 3kFx (Part 2/4)**
>
> >**Q2: “In Fig 5a, how is the gradient sparsity precisely defined?”**
>
> **Response:** Thank you for your comment. The gradient sparsity of the attention tensor is computed by first converting the gradient tensor to a floating-point NumPy array $ \\mathbf{G} $, then calculating the proportion of elements whose absolute values fall below a small threshold $ \\epsilon = 10^{-6} $. This metric quantifies the extent of near-zero gradients, which indicates sparse updates during training.
>
> The formula is as follows:
>
> $$ s = \\frac{1}{|\\mathbf{G}|} \\sum_{i,j} \\mathbb{I}(|\\mathbf{G}_{i,j}| < \\epsilon), $$
>
> where $s$ denotes the sparsity, $|\\mathbf{G}|$ is the total number of elements in $ \\mathbf{G} $, and $ \\mathbb{I} $ is the indicator function.
>
> ---
>
> >**Q3: “The framework retains the top 25% of sink heads, is this ratio empirically optimized, and how sensitive is the model performance to this threshold?”**
>
> **Response:** Thank you for raising this point. To address this, we conducted an ablation analysis on the fixed selection ratio for sink heads in **Appendix B.1**. Specifically, we evaluated the impact of different sink head retention strategies, with results summarized in the table below:
>
> | **Method** | **OVDEval↑** | **RefCOCO↑** | **GQA↑** |
> |------------|--------------|--------------|----------|
> | Base | 39.5 | 84.2 | 60.4 |
> | + SFT | 39.9 | 84.6 | 62.0 |
> | + 0% sink heads | 41.1 | 85.0 | 62.8 |
> | + 40% sink heads | 41.1 | 85.8 | 63.5 |
> | + 80% sink heads | 40.8 | 85.1 | 62.2 |
> | + 100% sink heads | 39.9 | 84.6 | 62.0 |
> | Ours(fixed 25% sink heads) | 42.8 | 86.1 | 63.5 |
> | **Ours** | **43.8** | **86.8** | **64.2** |
>
> We observe that retaining the top 25% of sink heads yields optimal performance across multiple benchmarks, with only mild sensitivity to the threshold.
>
> Furthermore, inspired by your suggestion, we performed an in-depth analysis of patterns within sink head subsets to refine this strategy, as detailed in the red-highlighted sections of **Sections 4.2 and 5**, along with detailed statistics, visualizations, and case studies in **Appendices B.4 and D**.
>
> We found that not all sink heads are equally beneficial: some disrupt effective modality fusion, marginalize visual cues, and bias outputs toward textual priors, ultimately degrading performance. To identify these, we compute cross-attention among all tokens within sink heads and observe distinct patterns in non-vision token cross-attention (illustrated in **Figure 5**): some heads exhibit uniformly dispersed attention across tokens (preserving global and contextual information), while others sink attention onto individual low-semantic tokens. Based on this insight, we propose a dynamic strategy that quantifies these patterns via the entropy of the non-vision token attention distribution, defined as follows.
>
> Let $\mathbf{A} \in \mathbb{R}^{L_q \times L_k}$ denote the attention matrix for a given head, where $L_q$ and $L_k$ are the query and key sequence lengths, respectively. Let $\mathcal{I}$ be the set of indices corresponding to non-vision tokens (keys). Extract the submatrix $\mathbf{A} _ \text{sub} = \mathbf{A}[:, \mathcal{I}] \in \mathbb{R}^{L _ q \times |\mathcal{I}|}$. Renormalize each row $i$ such that
> $$\mathbf{A} _ \text{sub}[i, :] \leftarrow \frac{\mathbf{A} _ \text{sub}[i, :]}{\sum _ {j \in \mathcal{I}} \mathbf{A} _ \text{sub}[i, j]},$$
> ensuring each row sums to 1 over non-image keys. Then, compute the average attention distribution $\mathbf{p} \in \mathbb{R}^{|\mathcal{I}|}$ where
> $$
> p _ j = \frac{1}{L _ q} \sum_{i=1}^{L _ q} \mathbf{A} _ \text{sub}[i, j]
> $$
> for each $j \in \mathcal{I}$. The entropy is given by
> $$
> H = -\sum_{j \in \mathcal{I}} p_j \log p_j.
> $$
>
> We define high-entropy heads as $sink_G$ heads and low-entropy heads as $sink_S$ heads. As shown in the table below on the OVDEval benchmark, adding an extra $sink_S$ head or ablating vision heads or $sink_G$ heads leads to significant performance drops, underscoring the harm of isolated token sinking and the value of visual cues and global context. Moreover, entropy values across heads follow a bimodal Gaussian distribution, enabling us to leverage its valley as a dynamic threshold for differentiating $sink_S$ and $sink_G$ heads and selectively removing the harmful ones, which further enhances our method's performance.
>
> | **Method**               | **Accuracy (%)** |
> |--------------------------|------------------|
> | Qwen2.5-VL-3B            | 39.5             |
> | w/o $sink_S$ head        | 43.8             |
> | w/ 1 $sink_S$ head       | 43.0             |
> | w/o 1 $sink_G$ head      | 43.2             |
> | w/o 1 vision head        | 42.6             |

---

> > ### Author Response · Authors · 2025-11-26
> > **Response to Reviewer 3kFx (Part 3/4)**
> >
> > >**Q4: “The tested models used are up to only 7B parameters. Would the same sparsification principle hold in larger models? Due to increased capacity of larger models, it's conceivable that the resinking problem would be alleviated.”**
> >
> > **Response:** Thank you for this insightful suggestion. Following your recommendation, we extended our experiments to larger models, including Qwen2.5VL-7B and Qwen2.5VL-32B, with results presented in **Tables 2 and 3** below.
> >
> > | Model                | VQA^v2 | GQA  | VQA^vg | MME    | MMB  | MMStar | AI2D | InfoVQA | TextVQA | DocVQA |
> > |----------------------|------------------|------|------------------|--------|------|--------|------|----------|----------|--------|
> > | LLaVA-1.5-7B         | 78.3             | 61.1 | 54.6             | 1808.4 | 61.1 | 33.2   | 55.7 | 41.2     | 64.7     | 69.4   |
> > | +SFT                 | 79.1             | 63.1 | 55.7             | 1899.6 | 61.9 | 34.5   | 56.2 | 43.7     | 65.5     | 71.2   |
> > | **+ Ours**           | **80.8**         | **65.2** | **58.5**     | **2018.8** | **63.2** | **36.3** | **57.1** | **46.3** | **67.5** | **74.4** |
> > | Qwen2.5VL-3B         | 76.7             | 60.4 | 54.3             | 2184.1 | 75.4 | 53.0   | 77.9 | 75.1     | 78.7     | 93.0   |
> > | +SFT                 | 77.9             | 62.0 | 55.2             | 2199.9 | 75.9 | 53.7   | 78.4 | 75.9     | 79.0     | 92.9   |
> > | **+ Ours**           | **79.7**         | **64.2** | **58.1**     | **2276.3** | **76.9** | **55.4** | **79.5** | **77.3** | **80.4** | **93.5** |
> > | InternVL2-2B         | 72.9             | 55.6 | 50.1             | 1864.3 | 69.1 | 48.9   | 73.1 | 58.8     | 73.4     | 86.4   |
> > | +SFT                 | 74.2             | 56.9 | 52.3             | 1899.1 | 70.0 | 49.6   | 73.9 | 59.1     | 73.8     | 86.6   |
> > | **+ Ours**           | **75.9**         | **59.0** | **55.4**     | **2006.5** | **71.6** | **50.8** | **75.7** | **60.5** | **75.9** | **88.2** |
> > | Qwen2.5VL-7B         | 81.6             | 65.8 | 60.5             | 2276.3 | 82.2 | 64.2   | 84.1 | 81.7     | 80.2     | 94.8   |
> > | +SFT                 | 81.9             | 66.1 | 61.0             | 2230.2 | 82.0 | 64.5   | 84.4 | 82.0     | 80.7     | 94.2   |
> > | **+ Ours**           | **82.6**         | **67.9** | **62.1**     | **2289.8** | **83.3** | **66.0** | **84.8** | **82.9** | **81.3** | **95.0** |
> > | Qwen2.5VL-32B        | 82.9             | 68.4 | 63.6             | 2297.4 | 83.8 | 70.3   | 85.2 | 83.4     | 82.8     | 94.8   |
> > | +SFT                 | 83.0             | 68.6 | 63.9             | 2255.4 | 83.8 | 69.6   | 85.1 | 83.0     | 82.9     | 94.4   |
> > | **+ Ours**           | **83.8**         | **69.9** | **64.5**     | **2326.6** | **84.5** | **71.3** | **85.7** | **83.8** | **83.9** | **95.1** |

---

> ### Author Response · Authors · 2025-11-26
> **Response to Reviewer 3kFx (Part 4/4)**
>
> | Model                | RefCOCO/+/g | LISA | RefGTA | OD^VG | OVDEval | MMVP | CVBench | CLEVER | CHAIR↓ | POPE↑ |
> |----------------------|-------------|------|--------|------------------|----------|------|----------|--------|-------------------|------------------|
> | LLaVA-1.5-7B         | 76.2        | 44.2 | 64.1   | 19.4             | 22.7     | 3.1  | 57.4     | 43.6   | 44.7              | 85.6             |
> | +SFT                 | 77.1        | 44.7 | 64.6   | 20.2             | 23.0     | 9.7  | 57.8     | 44.1   | 45.2              | 85.7             |
> | **+ Ours**           | **78.9**    | **50.1** | **66.2** | **24.8**       | **27.1** | **15.1** | **60.4** | **46.6** | **41.7**        | **86.4**         |
> | Qwen2.5-VL-3B        | 84.2        | 55.3 | 70.8   | 32.1             | 39.5     | 50.4 | 67.3     | 68.7   | 35.6              | 86.1             |
> | +SFT                 | 84.6        | 55.3 | 71.0   | 32.5             | 39.9     | 52.1 | 68.1     | 70.0   | 35.4              | 86.4             |
> | **+ Ours**           | **86.8**    | **58.1** | **72.9** | **36.7**       | **43.8** | **54.9** | **70.1** | **72.5** | **32.6**        | **87.4**         |
> | InternVL2-2B         | 77.8        | 46.1 | 66.4   | 21.7             | 24.9     | 39.6 | 56.5     | 57.1   | 37.8              | 86.2             |
> | +SFT                 | 78.1        | 45.6 | 66.9   | 23.2             | 25.3     | 40.4 | 57.2     | 57.9   | 37.9              | 86.0             |
> | **+ Ours**           | **80.1**    | **48.2** | **68.9** | **26.6**       | **29.9** | **42.7** | **59.2** | **59.6** | **34.3**        | **87.1**         |
> | Qwen2.5VL-7B         | 87.1        | 60.3 | 74.4   | 39.3             | 44.8     | 55.1 | 73.6     | 74.4   | 32.6              | 88.9             |
> | +SFT                 | 87.3        | 60.1 | 74.8   | 39.7             | 44.4     | 55.8 | 73.8     | 74.8   | 33.1              | 89.2             |
> | **+ Ours**           | **88.2**    | **63.6** | **76.0** | **42.1**       | **47.2** | **57.0** | **75.2** | **75.9** | **29.7**        | **89.6**         |
> | Qwen2.5VL-32B        | 89.8        | 65.9 | 77.5   | 43.1             | 49.3     | 60.4 | 77.2     | 78.5   | 28.2              | 90.3             |
> | +SFT                 | 89.7        | 63.3 | 77.8   | 43.3             | 49.3     | 60.8 | 77.1     | 78.7   | 30.1              | 90.1             |
> | **+ Ours**           | **90.6**    | **67.4** | **79.3** | **44.5**       | **51.8** | **62.5** | **79.0** | **80.0** | **21.2**        | **90.6**         |
>
> We observe that, when scaling to larger models, SFT yields minimal performance gains (particularly on vision-task benchmarks), whereas our method consistently delivers substantial improvements. This demonstrates the effectiveness, stability, and scalability of our sparsification principle. Additionally, we analyzed data scaling in **Figure 9**, revealing that SFT's benefits diminish as training data increases, with gains plateauing noticeably. In contrast, SADS exhibits more pronounced improvements, underscoring how the visual attention re-sinking phenomenon severely hampers existing multimodal models' ability to scale with data. Our approach effectively mitigates this issue, unlocking superior scaling potential.
>
> ---
>
> We sincerely thank you for your valuable suggestions, which have helped us refine and strengthen the paper. If you have any additional questions or concerns, please do not hesitate to let us know.

---

### Author Response · Authors · 2025-11-26
**General Response**

Dear AC and Reviewers,

We are deeply grateful for your dedication throughout the review and rebuttal phases, as well as for the constructive feedback that has significantly enhanced the clarity and refinement of our work. In response to your valuable suggestions, we have carefully revised the manuscript and highlighted all changes from the initial draft in **red** for your convenience.

In addition to addressing each reviewer's comments point-by-point below, we would like to summarize the key contributions of this work and highlight the new experimental results incorporated during the rebuttal phase.

---

We are pleased that the reviewers recognized and appreciated the following strengths and contributions:
1. "...developed a systematic approach to diagnose a commonly observed but not well-understood phenomenon. The analysis is logical, insightful and convincing..." **[Reviewer 3kFx]**
2. "...deeply identifies text-only supervision as the cause of suboptimal MLLM output layers...effectively addresses key issues to optimize output layers. Comprehensive experiments...validate its superiority." **[Reviewer 7PSW]**
3. "The observations...the illustrations...are all interesting. The finding...is interesting too. The performance gains are structural...The proposed method is effective and simple..." **[Reviewer VNAx]**

---

Inspired by the reviewers' insightful comments, we have incorporated the following key experiments and analyses during the discussion phase:
1. We conducted an ablation study on the proportion of retained sink heads in **Appendix B.1**. Furthermore, we provide in-depth analyses of sink heads in **Sections 4.2 and 5**, along with detailed statistics and case studies in **Appendices B.4 and D**. Specifically, we observed distinct patterns in the non-vision token cross-attention entropy: high-entropy $sink_G$ heads exhibit uniformly dispersed attention across tokens, preserving global and contextual information; in contrast, low-entropy $sink_S$ heads concentrate attention on individual low-semantic tokens, disrupting effective modality fusion, marginalizing visual cues, and biasing outputs toward textual priors, which ultimately degrades model performance. Additionally, we note that the entropy values across heads follow a bimodal Gaussian distribution, enabling us to use its valley as a dynamic threshold for differentiating $sink_S$ and $sink_G$ heads. This adaptive optimization further enhances our method's performance across a broad range of benchmarks.
2. We evaluated our method on scaled baselines (Qwen2.5VL-7B and Qwen2.5VL-32B) in **Tables 2 and 3**, and performed an ablation on scaling data size in **Figure 9**. Notably, when SFT struggles to improve performance, our method still achieves substantial gains, demonstrating its strong efficacy and scalability.
3. We ablated alternative training objectives, including adding regularization terms to upweight vision heads and reweighting attention for sink heads, in **Appendix B.3**. Our approach consistently outperforms these alternatives, confirming that it fundamentally addresses the output suboptimality caused by sparse attention gradients.
4. We demonstrate in **Appendix B.2** that early stopping during training cannot effectively resolve output suboptimality and may instead reduce model performance due to underfitting.
5. We provide additional qualitative results in **Appendix D** to visually demonstrate the enhancements introduced by our method.

---

Finally, we sincerely thank you for taking the time to provide constructive feedback on our work. Should you have any further questions or concerns, please let us know, we are happy to address them.

Best,
Authors

---

> ### Author Response · Authors · 2025-12-01
> **Further Updates After the Discussion Phase**
>
> We are pleased to have received **Reviewer VNAx's** response to our rebuttal: **"I am satisfied with the detailed explanations, clarifications, and paper modifications. I will increase my score."** Unfortunately, we did not receive responses from **Reviewers 3kFx** and **7PSW**. The resolved issues are summarized below:
>
> > **Alternative Training Objectives for Gradient Sparsity (Reviewer 3kFx)**
>
> We explored regularization and attention reweighting in **Appendix B.3**, with results showing modest gains over SFT. However, SADS outperforms them by fundamentally addressing re-sinking, demonstrating the effectiveness of our method.
>
> > **Early Checkpoints Performance (Reviewer 3kFx)**
>
> We added experiments in **Appendix B.2** comparing SFT and our method across iteration steps. Results reveal that early stopping causes underfitting and degradation, while our method consistently outperforms SFT at all checkpoints, proving its stability.
>
> > **Sink Heads Retention Ratio and Dynamic Strategy (Reviewers 3kFx and 7PSW)**
>
> We conducted ablations on fixed ratios in **Appendix B.1**, finding 25% optimal with mild sensitivity. Further, we analyzed sink head patterns in **Sections 4.2/5** and **Appendices B.4/D**, distinguishing high-entropy $sink_G$ (preserving global and contextual information) from low-entropy $sink_S$ (disrupting modality fusion) via non-vision token cross-attention entropy.  Moreover, entropy values across heads follow a bimodal Gaussian distribution, enabling us to leverage its valley as a dynamic threshold for differentiating $sink_S$ and $sink_G$ heads and selectively removing the harmful ones, which further enhances our method's performance.
>
> > **Scaling to Larger Models and Data (Reviewers 3kFx and 7PSW)**
>
> We extended evaluations to Qwen2.5VL-7B/32B in **Tables 2/3**, where SFT shows minimal gains on vision tasks, but our method delivers substantial improvements. Data scaling ablations in **Figure 9** highlight re-sinking's hindrance to SFT, making it difficult to improve performance as data volume increases, while SADS unlocks better scalability.
>
> > **Quantifying Re-sinking Impact (Reviewer 7PSW)**
>
> We revisited **Figure 2** by applying targeted fine-tuning to output heads across layers and incorporating the projected performance in the absence of re-sinking, which illustrates the severity of the performance limitations imposed by this phenomenon. **Tables 2/3** show that our method's gains (e.g., +5.9 on LISA for LLaVA-1.5-7B) far exceed SFT's minimal or even negative changes, especially on vision-centric tasks.

---

### Meta-Review · Area_Chair_C3Sa · 2025-12-18

**Summary:**

This paper addresses the issue of Visual Attention Re-sinking and proposes a parameter-free method called Sink Attention Dynamic Sparsification (SADS). Through a series of experiments, the authors demonstrate the effectiveness of the method across multiple benchmarks, showing improvements in model performance. In response to reviewer comments, the authors expanded experiments to include different model sizes, validating the method's effectiveness on large-scale models. They also conducted an ablation study on the selection ratio of "sink heads," proving the robustness of this strategy. Additionally, the identification of unhealthy "sink heads" and their impact on performance was clarified. These responses have made the paper clearer and the experiments more comprehensive. The paper provides a valuable contribution to addressing the poor performance of output layers in MLLMs, with experimental validation supporting the proposed method's effectiveness. Thus, the paper is recommended for acceptance.

**Reviewer Concerns:**

Almost all reviewer concerns were addressed, summarized as follows:

•  Reviewer 3kFx: The authors provided additional experimental results to address the concern regarding gradient sparsity and training checkpoints, demonstrating the stability of the SADS method across different training steps and comparing it with other methods.

•  Reviewer 7PSW: Regarding the effectiveness of the fixed 25% retention ratio for sink heads, the authors performed ablation studies showing its optimal performance across multiple benchmarks and also validated the method on larger models, addressing concerns about the method's generalizability.

•  Reviewer VNAx: For concerns about sink head identification and its impact, the authors provided more detailed quantitative analysis, particularly by optimizing sink head selection with an adaptive threshold strategy, improving model performance.

**Reviewer Scores:**

I believe that if each reviewer had actively participated in the discussion, they would likely have made positive changes to their scores.

•  Reviewer 3kFx: Given the additional experiments and responses, the reviewer would likely increase their score, particularly in "contribution" and "experiments," as the paper now provides more in-depth analysis and solutions to the gradient sparsity issue.

•  Reviewer 7PSW: The reviewer would likely raise their score as well, especially given the additional validation of the method across different model sizes and benchmarks, which addresses their concerns regarding scalability and method robustness.

•  Reviewer VNAx: The reviewer has already increased their score due to the added clarity on sink head identification and the adaptive strategy to optimize their selection, which resolves some of their major concerns.

---

### Decision · Program_Chairs · 2026-01-26

Accept (Poster)